# Rethinking Global Text Conditioning in Diffusion Transformers

**Nikita Starodubcev**[1*] **Daniil Pakhomov**[2] **Zongze Wu**[2] **Ilya Drobyshevskiy**[1]
**Yuchen Liu**[2] **Zhonghao Wang**[2] **Yuqian Zhou**[2] **Zhe Lin**[2] **Dmitry Baranchuk**[1]

[1]Yandex Research, [2]Adobe Research
https://github.com/quickjkee/modulation-guidance

## Abstract

Diffusion transformers typically incorporate textual information via (i) attention layers and (ii) a modulation mechanism using a pooled text embedding. Nevertheless, recent approaches discard modulation-based text conditioning and rely exclusively on attention. In this paper, we address **whether modulation-based text conditioning is necessary and whether it can provide any performance advantage**. Our analysis shows that, in its conventional usage, the pooled embedding contributes little to overall performance, suggesting that attention alone is generally sufficient for faithfully propagating prompt information. However, we reveal that the pooled embedding can provide significant gains when used from a different perspective—serving as guidance and enabling controllable shifts toward more desirable properties. This approach is training-free, simple to implement, incurs negligible runtime overhead, and can be applied to various diffusion models, bringing improvements across diverse tasks, including text-to-image/video generation and image editing.

## 1 Introduction

Since the pioneering works on diffusion models (DMs) (Ho et al., 2020; Song et al., 2020), the UNet architecture (Ronneberger et al., 2015) has served as the dominant backbone for diffusion-based image generation. This trend extended to text-to-image models (Saharia et al., 2022; Nichol et al., 2021), which employ UNet-based architectures (Rombach et al., 2022) and incorporate the CLIP text encoder (Radford et al., 2021) to condition the model on text sequences through the attention mechanism (Vaswani et al., 2017). Later, models such as Podell et al. (2023) began to incorporate the pooled CLIP embedding via modulation mechanisms (Karras et al., 2017; 2019), in addition to the token-wise text embeddings. More recently, works including Labs et al. (2025); Labs (2024); Esser et al. (2024); Kong et al. (2024); Cai et al. (2025) have adopted transformer-based architectures (Peebles & Xie, 2023) while retaining modulation-based text conditioning. Recent models (Wan et al., 2025; Wu et al., 2025; Agarwal et al., 2025; Xie et al., 2024) discard global text conditioning, achieving comparable text alignment by relying solely on attention. This transition raises questions about the role and necessity of global text conditioning, which we aim to explore.

We observe that, at first glance, modulation-based text conditioning appears non-contributory, and attention alone is sufficient to capture textual information. However, we argue that it is premature to discard global text conditioning and that it should instead be leveraged from a different perspective. Specifically, we draw inspiration from the interpretability of the modulation mechanism (Karras et al., 2019) and the ability of CLIP to control it (Garibi et al., 2025). We suggest that the pooled text embedding can act as a corrector, adjusting the diffusion trajectory toward better modes.

In summary, our contributions are as follows: **(1)** We conduct an in-depth analysis of global text conditioning in contemporary DMs and find that it plays only a minor role relative to attention-based text conditioning. **(2)** We show that global text conditioning can yield significant improvements when viewed from the perspective of *modulation guidance*. Furthermore, we enhance its effectiveness by proposing dynamic strategies. **(3)** We introduce techniques for integrating the pooled embedding

---

*Work partially done during an internship at Adobe Research

into fully attention-based models, thereby improving their performance via modulation guidance. **(4)** From a practical standpoint, our approach is simple to implement, incurs negligible overhead, and delivers performance gains on state-of-the-art multi- and few-step DMs across text-to-image/video and image-editing tasks.

## 2 RELATED WORK

Several post-training approaches have been proposed to improve DM quality. The first group centers on *classifier-free guidance (CFG) modifications* (Ho & Salimans, 2022). Specifically, prior works improve CFG by optimizing scale factors (Fan et al., 2025), addressing off-manifold challenges (Chung et al., 2024), modifying the unconditional branch (Karras et al., 2024), mitigating oversaturation at high CFG scales (Sadat et al., 2024; 2025; Lin et al., 2024), and introducing dynamic CFG strategies (Kynkäänniemi et al., 2024; Sadat et al., 2023; Wang et al., 2024; Yehezkel et al., 2025). In contrast, our method complements CFG and, importantly, can also be applied to few-step DMs (Song et al., 2023; Sauer et al., 2024b; Yin et al., 2024a; Starodubcev et al., 2025) that do not use CFG.

The second group focuses on *test-time optimization*. A dominant line of work (Chefer et al., 2023; Seo et al., 2025; Yiflach et al., 2025; Li et al., 2023; Rassin et al., 2023; Agarwal et al., 2023; Dahary et al., 2024; Marioriyad et al., 2025; Binyamin et al., 2025; Phung et al., 2024; Chen et al., 2024; Kwon et al., 2022) relies on handcrafted loss functions, typically guided by heuristics about how attention maps should behave, and optimizes these maps accordingly. Other methods focus on optimizing only the initial noise rather than the full denoising trajectory (Eyring et al., 2025; Ma et al., 2025a; Eyring et al., 2024; Guo et al., 2024), or on fine-tuning LoRAs to extract different concepts (Gandikota et al., 2024). In contrast, our approach avoids complex loss design and intensive model tuning while still improving performance.

Finally, works most closely related to ours are *attention guidance methods*. These methods (Chen et al., 2025; Hong et al., 2023; Ahn et al., 2025; Nguyen et al., 2024) leverage positive and negative prompts, compute attention outputs for both, and perform controlled extrapolation in the attention space—pushing the model toward positive prompts and away from negative ones. Our approach also relies on guidance in feature space but applies it through a small MLP rather than through attention.

## 3 MODULATION LAYERS

In this section, we briefly recap the key components of modulation layers used in transformer DMs.

State-of-the-art text-to-image DMs (Labs, 2024; Cai et al., 2025) typically represent images as sequences of continuous tokens, aligning them with text tokens in a unified representation. This combined sequence is processed through a series of transformer blocks (Peebles & Xie, 2023), which primarily consist of MLPs, normalization, and attention layers. To condition the model on a text prompt, two types of encoders are usually used: a T5 (Raffel et al., 2020) and a CLIP text encoder (Radford et al., 2021), which operate as follows:

$$\mathbf{y}(\mathbf{p}, t) = \mathrm{MLP}\big(t,\ \mathrm{CLIP}(\mathbf{p})\big), \quad \mathbf{s} = \big[\mathrm{T5}(\mathbf{p}),\ \mathbf{x}\big], \tag{1}$$

Here, $\mathbf{y}$ denotes a global conditioning vector derived from the time step $t$ and the pooled embedding of the prompt $\mathbf{p}$, whereas $\mathbf{s}$ denotes the concatenated sequence of image tokens $\mathbf{x}$ and text tokens $\mathrm{T5}(\mathbf{p})$. The sequence $\mathbf{s}$ is then processed via cross-attention to incorporate text information, while the global conditioning vector $\mathbf{y}$ is shared across the entire model and constructs a modulation space that influences the modulation layers.

$$\mathrm{Mod}(\mathbf{s},\ \mathbf{y}) = \alpha_{\mathbf{s}}(\mathbf{y}) \cdot \mathbf{s} + \beta_{\mathbf{s}}(\mathbf{y}), \tag{2}$$

Here, $\alpha_{\mathbf{s}}$ and $\beta_{\mathbf{s}}$ are the coefficients of the modulation layer, representing scaling and shifting operations, respectively. Notably, modulation layers have proved effective in enabling semantic control and transformation in GANs (Karras et al., 2019; 2020; 2021). In DMs, they have been used to address image editing problems (Garibi et al., 2025; Dalva et al., 2024). While these layers have shown effectiveness in semantic control tasks, their role in improving image generation quality remains unexplored.

| Configuration | | CLIP Score ↑ | PickScore ↑ | ImageReward ↑ |
|---|---|---|---|---|
| **FLUX schnell** | | | | |
| Initial, | short | 30.1 | 21.6 | 6.2 |
| w/o CLIP, | short | 29.0 (−1.1) | 21.3 (−0.3) | 4.5 (−1.7) |
| w/o T5, | short | 28.9 (−1.2) | 21.0 (−0.6) | 1.5 (−4.7) |
| Initial, | long | 33.1 | 21.0 | 10.3 |
| w/o CLIP, | long | 32.8 (−0.3) | 21.0 (−0.0) | 10.4 (+0.1) |
| w/o T5, | long | 30.7 (−2.4) | 19.9 (−1.1) | 2.4 (−7.9) |
| **HiDream-Fast** | | | | |
| Initial, | short | 30.3 | 21.8 | 7.9 |
| w/o CLIP, | short | 30.3 (−0.0) | 21.8 (−0.0) | 8.1 (+0.1) |
| w/o Llama, | short | 20.2 (−10.1) | 18.2 (−3.6) | −21.5 (−29.4) |
| Initial, | long | 32.9 | 21.5 | 12.8 |
| w/o CLIP, | long | 32.9 (−0.0) | 21.5 (−0.0) | 13.0 (+0.2) |
| w/o Llama, | long | 16.8 (−16.1) | 16.0 (−4.5) | −20.8 (−33.6) |

Table 1: Image quality results for short and long prompts. The CLIP embedding does not affect output quality on long prompts for **FLUX schnell** and has no effect for **HiDream-Fast**.

Figure 1: **(top)** Difference between images (Dream-Sim) with and without CLIP as a function of prompt length. **(bot)** For long prompts, images without CLIP generally do not differ from the initial ones.

## 4 ANALYSIS OF THE POOLED TEXT EMBEDDING ROLE

In recent DMs, there is a trend to discard the pooled text embedding and rely solely on the timestep $t$ to produce $\mathbf{y}$, i.e., $\mathrm{MLP}\big(t, \mathrm{CLIP}(\mathbf{p})\big) \to \mathrm{MLP}(t)$. In this setup, the text is incorporated only through the text encoder T5. However, no strict justification for this design choice has been provided. Therefore, in this section, we investigate the impact of the pooled embedding on the generative performance of DMs.

**Influence of the CLIP pooled embedding.** First, we analyze the influence of CLIP on text-to-image generation performance. To this end, we examine two contemporary models: FLUX schnell and HiDream-Fast. Specifically, we analyze the impact of CLIP by removing the pooled embedding, setting $\mathrm{CLIP}(\mathbf{p}) \to 0$, and comparing it to the standard case with CLIP enabled. Our key observation is that **the pooled CLIP embedding is partially inactive in FLUX schnell and fully inactive in HiDream-Fast**.

Specifically, we find that the influence of CLIP in **FLUX schnell** is inconsistent: it is negligible for long prompts but can be impactful for short ones. To confirm this, we construct two subsets of prompts (1K each) from the MJHQ dataset (Li et al., 2024): short (10 tokens) and long (77 tokens). We then evaluate the DM's performance on each subset. In Table 1 (top), we report image quality metrics (CLIP Score, PickScore, and ImageReward) for each subset. We observe that for long prompts, CLIP has little effect, with only a minimal impact on quality. In contrast, for short prompts, its influence is more pronounced.

Moreover, in Figure 1, we analyze the difference between images generated with and without CLIP as a function of prompt length (measured by the number of tokens). We find that for longer prompts, the deviation from the initial generation becomes negligible, and the images fully resemble the initial ones, as visually confirmed in Figure 1 (bottom).

For **HiDream-Fast**, we observe slightly different behavior: the CLIP pooled embedding exhibits no effect for either short or long prompts, as numerically confirmed in Table 1 (bottom).

**Influence of the pooled embedding on other models.** Additionally, we explore the reintegration of CLIP into a DM from which it was originally absent. To this end, we consider the COSMOS model (Agarwal et al., 2025) and incorporate the CLIP pooled embedding into it as described in Section 5. In this case, we observe the same behavior as with the HiDream-Fast model: CLIP has no influence. This result is numerically confirmed in Section 6. Finally, in Appendix A, we observe the same effect in the instruction-guided image editing task performed with the FLUX Kontext model. In Section 6, we show that this limitation can result in insufficient editing strength for complex cases.

## 5  MODULATION GUIDANCE

Our observations raise questions about the necessity of using the pooled embedding in generative tasks. **However, although the pooled text embedding may seem uninformative in some cases, we propose reconsidering its role from a different perspective—one that can lead to improved generative performance in DMs.**

**Guidance in modulation space.** We draw inspiration from the understanding that modulation layers can drive semantic changes in generated images (Karras et al., 2019). Moreover, the CLIP encoder is trained to construct a shared space between images and text, resulting in interpretable geometry. Thus, we suggest that CLIP enables interpretable modifications of the modulation space using natural language and guides the model toward modes with more desirable properties.

We propose a training-free, plug-and-play technique to reactivate CLIP and strengthen its influence during generation, drawing inspiration from Garibi et al. (2025). Specifically, we amplify its effect by introducing guidance in the modulation space.

$$\mathbf{y}(\mathbf{p}, t) \rightarrow \hat{\mathbf{y}}(\mathbf{p}, \mathbf{p}_+, \mathbf{p}_-, t) = \mathbf{y}(\mathbf{p}, t) + w \cdot \big(\mathbf{y}(\mathbf{p}_+, t) - \mathbf{y}(\mathbf{p}_-, t)\big). \tag{3}$$

We note that $\hat{\mathbf{y}}$ affects only the modulation coefficients and is shared across all DM blocks, thereby incurring negligible computational overhead compared to basic generation. Moreover, this technique can be applied on top of CFG guidance or with distilled DMs that do not rely on CFG.

To provide intuition behind the guidance, we first analyze it from the perspective of semantic changes. Prior work has focused on identifying interpretable directions in DMs, either through supervised (Gandikota et al., 2024) or unsupervised approaches (Gandikota et al., 2025). In contrast, we demonstrate that such interpretable directions are already embedded within the model and can be accessed by shifting in the modulation space. Specifically, in Figure 2, we consider two examples: $\mathbf{p}_+ = $ Long hair; Modern car and $\mathbf{p}_- = $ Short hair; Old car. We observe that the pooled embedding can substantially influence the generated image, leading to both local (hair length) and global (car style) changes.

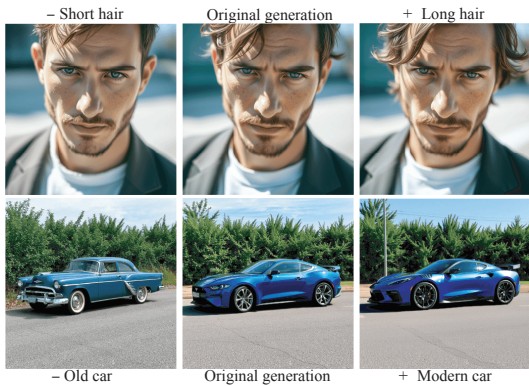

Figure 2: The modulation guidance enables local (top) and global (bottom) changes and encourages its use to shift a DM toward modes with better properties.

Our observations suggest that modulation guidance provides an additional degree of freedom in generation, beyond what CFG offers. Building on this, we propose using it to enhance generation quality across multiple dimensions. Specifically, we consider **general changes:** `aesthetics`, `complexity`, and **specific changes:** `hands correction, object counting, color, position`. For the latter, we focus on common criteria typically measured in T2I benchmarks (Ghosh et al., 2023). Notably, our technique requires only the selection of a suitable prompt for each category—no additional training or fine-tuning is necessary. In Appendix D, we present the prompts used for each targeted aspect.

**Dynamic modulation guidance.** We find that a constant guidance scale $w$ is generally effective, but excessively high values can overweight the prompt and cause the model to neglect textual information (Appendix C). To address this, we draw inspiration from dynamic CFG (Sadat et al., 2023; Kynkäänniemi et al., 2024), which has shown promising results in DMs. Unlike dynamic CFG, we aim to adjust $w$ across layers rather than across time steps.

We consider the simplest variant present in Figure 3(b). We discuss more strategies in Appendix B. First, we compare the dynamic version against constant modulation guidance in terms of the aesthetics–prompt fidelity trade-off. To this end, we apply both types of guidance with different scales $w$ on 1K prompts from the MJHQ dataset (Lian et al., 2023). We compute PickScore (Kirstain et al., 2023) for aesthetics quality and CLIP score (Hessel et al., 2021) for text correspondence. The results presented in Figure 3(a) confirm that dynamic guidance provides a better trade-off than constant

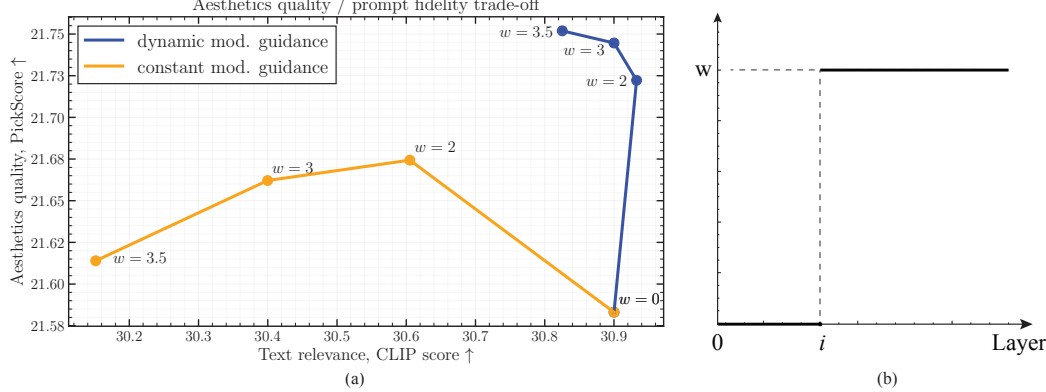

Figure 3: **Analysis on dynamic modulation guidance.** (a) Dynamic guidance offers a better trade-off between aesthetic quality and prompt correspondence than constant modulation guidance. (b) We use a step function, controlled by $i$, that skips the first few layers of the model as our form of dynamic guidance. Additional variants are explored in Appendix B.

guidance. Our approach improves image quality without compromising prompt correspondence relative to $w = 0$ (the initial model without modulation guidance).

In our experiments, we find that dynamic modulation guidance generalizes well across tasks, suggesting that it can be applied to new tasks without additional tuning. We also observe that more complex strategies (Appendix C) can yield better results in some cases, offering an additional degree of improvement.

**What does modulation guidance actually do?**
We address the question of how the model is affected by the guidance in improving the generated content. To this end, we analyze the case of `hands correction`. Specifically, in Figure 4(a), we visualize the attention map corresponding to the word `hands` for a specific image. Interestingly, we observe that the model places greater focus on the relevant region, highlighting it more distinctly. In addition, in Figure 4(b, left), we plot the averaged attention map for all tokens in the corresponding prompt. We find that the model primarily shifts its attention toward more relevant tokens—such as `hands` and `child`.

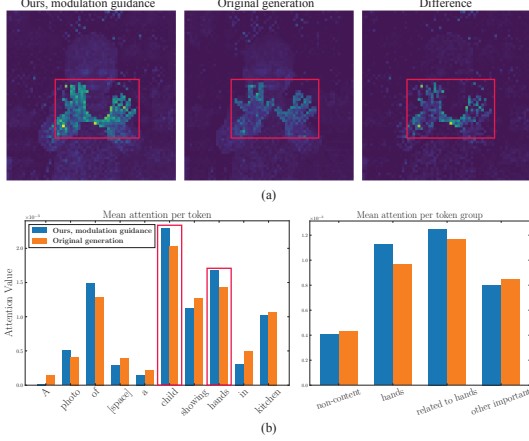

Figure 4: After applying modulation guidance, the model focuses more on the desired features, such as hands (a, b).

To confirm this intuition, we analyze a subset of prompts focused on `hands correction` and split all tokens into four groups: non-content tokens, the token `hands`, tokens related to hands, and other important tokens. The results in Figure 4(b, right) confirm that the model shifts its attention toward `hands` and hand-related tokens.

**Integrating the pooled text embedding into CLIP-free models.** Finally, we extend modulation guidance to models without pooled text embeddings, showing that it can improve generation quality. To this end, we fine-tune existing text-to-image/video models Agarwal et al. (2025); Wan et al. (2025) by introducing the pooled embedding. Specifically, we train a small MLP on top of the pooled text embedding and add it to the timestep embedding, while keeping the rest of the network frozen. The model behaves identically to the original when the pooled embedding is set to 0. Importantly, we train on the model's own synthetic data to ensure that improvements do not stem from dataset differences.

Table 2: Performance of text-to-image DMs with and without modulation guidance (gray) on Aesthetics and Complexity, evaluated with human preferences and automatic metrics. Human win rates are reported with respect to the original model; green indicates statistically significant improvement, red a decline. For automatic metrics, **bold** denotes improvement over the original model.

| Model | Side-by-Side Win Rate, % | | | | Automatic Metrics, COCO 5k | | | |
|---|---|---|---|---|---|---|---|---|
| | Relevance ↑ | Aesthetics ↑ | Complexity ↑ | Defects ↑ | PickScore ↑ | CLIP ↑ | IR ↑ | HPSv3 ↑ |
| **FLUX schnell** | | | | | 22.9 | 35.6 | 10.2 | 11.3 |
| Aesthetics | 48 | **72** | **78** | 48 | **23.1** | **35.8** | **11.0** | **11.8** |
| Complexity | 53 | **56** | **69** | 47 | **23.0** | **35.9** | **10.8** | **11.4** |
| **FLUX dev** | | | | | 23.1 | 34.7 | 10.5 | 12.4 |
| Aesthetics | 44 | **56** | **69** | 52 | **23.2** | 34.5 | **11.0** | **12.8** |
| Complexity | 48 | **59** | **72** | 47 | 23.1 | 34.6 | **11.1** | **12.8** |
| **SD3.5 Large** | | | | | 23.0 | 35.8 | 10.5 | 11.1 |
| Aesthetics | 50 | **62** | **70** | 47 | **23.1** | **35.9** | **10.7** | **11.2** |
| Complexity | 49 | 49 | **60** | 45 | 23.0 | 35.8 | **11.7** | 11.0 |
| **HiDream** | | | | | 23.4 | 34.4 | 11.7 | 13.2 |
| Aesthetics | 49 | **60** | **80** | 46 | **23.5** | 34.4 | **12.1** | **13.7** |
| Complexity | 47 | 52 | **70** | 45 | **23.5** | 34.4 | **11.9** | **13.3** |
| **COSMOS** | | | | | 23.0 | 35.0 | 11.4 | 12.3 |
| + CLIP | 50 | 49 | 43 | 50 | 23.0 | 35.0 | 11.4 | 12.2 |
| Aesthetics | 50 | **60** | **70** | 45 | **23.2** | 35.0 | **11.7** | **12.6** |
| Complexity | 50 | 52 | **61** | 44 | 23.0 | **35.4** | **11.8** | **12.4** |

We highlight two important aspects of the training process. First, we propagate the textual prompt solely through the pooled text embedding, using an unconditional prompt for T5. This design forces the model to perceive textual information through the pooled embedding. Second, we employ a distillation-based training regime. Specifically, we sample a clean image, add noise to it, and then generate two predictions: one from the original model (without the pooled embedding) and one from the modified model (with the pooled embedding). The objective is to minimize the MSE loss between these two predictions. This distillation approach is well-suited for few-step DMs, as it eliminates the need for complex adversarial or distribution-matching losses (Yin et al., 2024a).

## 6 EXPERIMENTS

### 6.1 TEXT-TO-IMAGE GENERATION

**Configuration.** We validate our approach on state-of-the-art text-to-image DMs that include modulation-based text conditioning: FLUX schnell (Sauer et al., 2024a), FLUX (Labs, 2024), SD3.5 Large (Esser et al., 2024), and HiDream (Cai et al., 2025). In addition, we consider the CLIP-free COSMOS model (Agarwal et al., 2025) and fine-tune it for 4K iterations to introduce the pooled text embedding. We train the model on its own synthetic dataset of 500K samples, following the generation settings of Agarwal et al. (2025) and using prompts from Li et al. (2024).

We evaluate performance using two types of metrics: human preference and automatic evaluation. Human preference is measured via side-by-side comparisons, where annotators assess image pairs on four criteria: text relevance, aesthetics, complexity, and defects (details in Appendix J). For general changes, we use 128 prompts from PartiPrompts (Yu et al., 2022), generating two images per prompt. For specific changes, we use 70 prompts from CompBench (Jia et al., 2025) for `object counting` and 200 LLM-generated prompts for `hands correction`. For automatic evaluation, we report CLIP score (Hessel et al., 2021), ImageReward (IR) (Xu et al., 2023), PickScore (PS) (Kirstain et al., 2023), and HPSv3 (Ma et al., 2025b), tested on 5K prompts from COCO2014 (Lin et al., 2014). We also use GenEval (Ghosh et al., 2023) to validate modulation guidance across multiple benchmark criteria.

Our main baselines are the original models without modulation guidance. In addition, we consider the Normalized Attention Guidance approach (Chen et al., 2025) and LLM-enhanced prompt modifiers (Lian et al., 2023). Finally, we include the test-time optimization method Concept Sliders (Gandikota et al., 2024) for the `hands correction` task.

FLUX, schnell  FLUX  Hi-Dream  SD3.5 Large  COSMOS

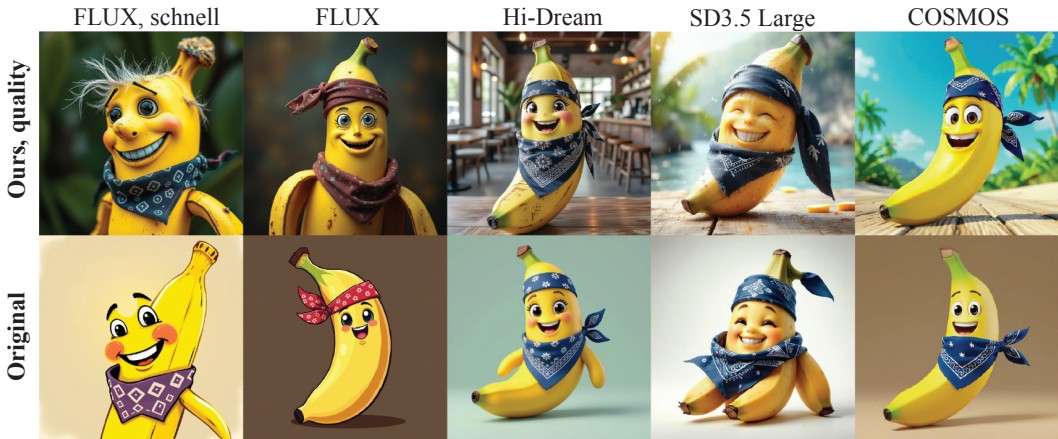

*a smiling banana wearing a bandana*

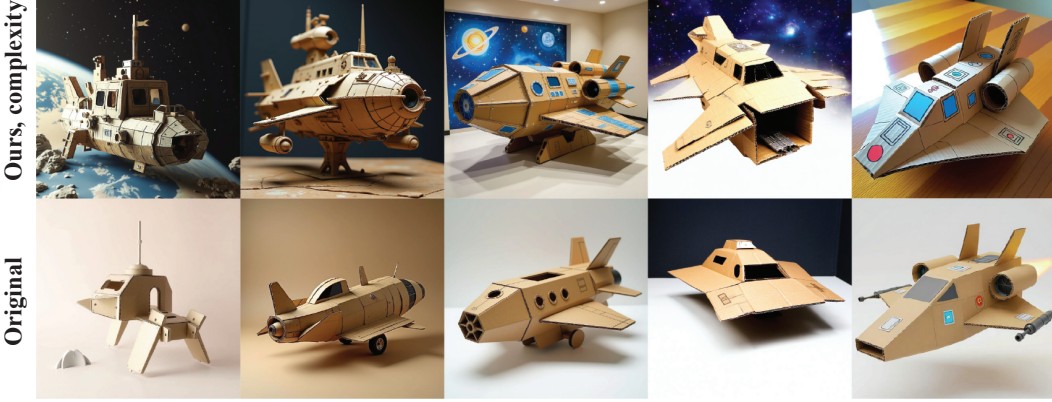

*a cardboard spaceship*

Figure 5: Qualitative results of modulation guidance for `Aesthetics` (top) and `Complexity` (bottom). The `Aesthetics` guidance notably improves image quality, while the `Complexity` guidance can enhance the complexity of both the main object and background details.

**General changes.** In this case, we focus on two aspects for improvement: `aesthetics` and `complexity`. These aspects are crucial for text-to-image generation and are typically the targets of self-supervised fine-tuning techniques (Startsev et al., 2025) or RL-based approaches (Wallace et al., 2024), which are commonly adopted in DMs. However, we demonstrate that our simple technique achieves significant improvements without any fine-tuning. The only requirement is to select appropriate positive and negative prompts, along with a suitable dynamic guidance strategy. Our choices are summarized in Table 5 and discussed in Appendix D.

Table 2 reports numerical results, showing clear human preference gains for both aspects. `Aesthetics` guidance significantly improves both aesthetics and complexity, while `complexity` guidance mainly enhances complexity. Automatic metrics show consistent ImageReward gains across all models and HPSv3 improvements in most cases, except for SD3.5 Large with `complexity` guidance. Importantly, we observe that introducing CLIP into COSMOS does not improve performance and even reduces complexity; gains appear only when combined with modulation guidance, confirming that CLIP alone is ineffective. We note slight drops in text relevance for FLUX dev and in defects for COSMOS, though these are minor. Qualitative examples are shown in Figure 5 and Appendix I.

**Specific changes.** Next, we focus on improving `object counting`, `hands correction`, `color`, and `position`. The first two are particularly important, as they have been extensively studied in prior work (Binyamin et al., 2025; Gandikota et al., 2024). For `object counting`, we use the number of target objects as the positive direction, while for `hands correction`, we draw

Table 3: Quantitative results of the modulation guidance for specific changes. The modulation guidance yields improvements according to GenEval and human preference.

| Model | | GenEval | | | SbS Win Rate, % | |
|---|---|---|---|---|---|---|
| | | Object Counting | Color | Position | Object Counting | Hands correction |
| **FLUX schnell** | Original | 56 | 79 | 25 | 39 | 41 |
| | Ours | 65 (+9) | 86 (+7) | 30 (+5) | 61 (+22) | 59 (+18) |

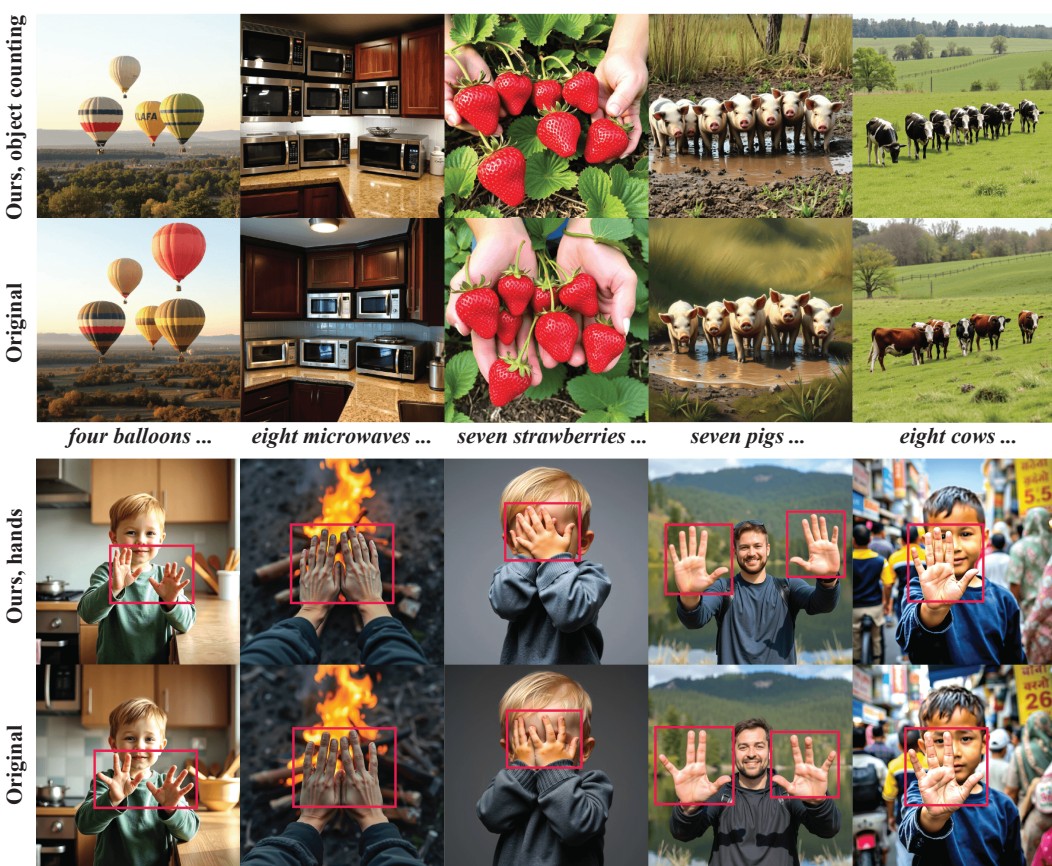

Figure 6: Qualitative results of the modulation guidance for `Object counting` (top) and `Hands correction` (bottom).

inspiration from Gandikota et al. (2024) in designing positive and negative prompts. Further details are provided in Table 5.

We present the results in Table 3 and Figure 6. Improvements are observed in several aspects of the GenEval benchmark, including object counting, color, and position. According to human evaluation, our approach improves the original model by 22% in `object counting` and 18% in `hands correction`. We report text relevance and defects as the evaluation criteria for object counting and hands correction, respectively.

**Comparison with baselines.** Normalized Attention Guidance (Chen et al., 2025) targets general changes, so we compare it with our `aesthetics` guidance using SbS evaluation. Similarly, we compare Concept Sliders (Gandikota et al., 2024) with our `hands correction` guidance by evaluating defects. For LLM-enhanced prompts (Lian et al., 2023), we consider general changes, `hands correction`, and `object counting`. Results in Appendix E (Tables 8 and 9) show that our approach outperforms Normalized Attention Guidance by 34% and Concept Sliders by 16%, without additional computational overhead. Moreover, Table 8 shows that modulation guidance can further improve performance when combined with LLM-enhanced prompts.

Table 4: Quantative evaluation on VBench. The results show an improved dynamic degree compared to the original models and baseline approach (normalized attention guidance).

| Model, video | | total score ↑ | motion smoothness ↑ | dynamic degree ↑ | aesthetic quality ↑ | overall consistency ↑ |
|---|---|---|---|---|---|---|
| Hunyan, 13B | Original | 56.68 | **99.23** | 50.51 | 55.88 | 21.08 |
| | Modulation guidance | **57.56** | 99.03 | **53.61** | **56.50** | **21.09** |
| CausVid, 1.3B | Original | 62.72 | **98.76** | 75.25 | 57.85 | 19.01 |
| | + CLIP | 62.82 | 98.63 | 76.38 | 57.77 | 18.49 |
| | Norm. attent. guidance | 63.58 | 98.39 | 74.22 | **62.08** | **19.61** |
| | Modulation guidance | **65.43** | 98.45 | **86.59** | 57.65 | 19.02 |

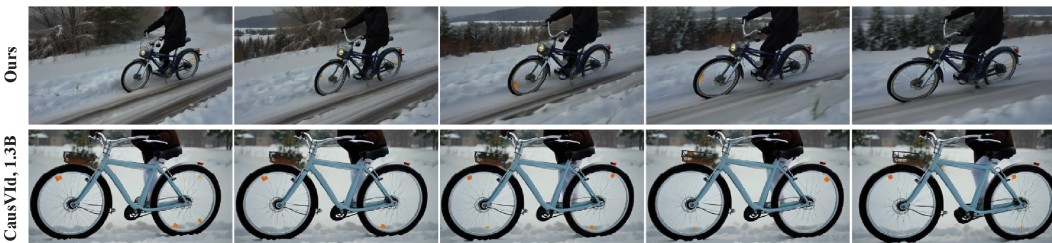

*A bicycle gliding through a snowy field.*

Figure 7: Qualitative comparison between the original CausVid and CausVid with modulation guidance.

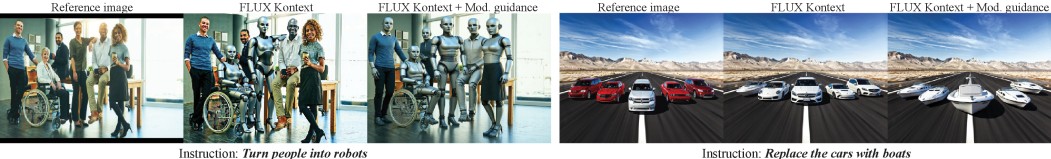

Figure 8: Qualitative results for text-guided image editing tasks. We observe that FLUX Kontext sometimes struggles with complex edits, while modulation guidance can mitigate this limitation.

## 6.2 TEXT-TO-VIDEO GENERATION

**Configuration.** We apply modulation guidance to Hunyuan 13B (Kong et al., 2024) and CausVid 1.3B (Yin et al., 2024b). The latter does not include a CLIP model, so we fine-tune it for 1K iterations. To evaluate performance, we use VBench (Huang et al., 2024), which covers various aspects. In this experiment, we apply the same `aesthetics` guidance as in the text-to-image task. In addition, we compare our approach with Normalized Attention Guidance.

**Results.** The results are presented in Table 4 and Figure 7. Importantly, we observe improvements in dynamic degree for both models, with particularly strong gains for CausVid. This is notable because CausVid is distilled from WAN (Wan et al., 2025), and video models typically lose dynamics after distillation. Furthermore, we find that incorporating CLIP provides no improvement. Additional visual comparisons are provided in Appendix I.

## 6.3 INSTRUCTION-GUIDED IMAGE EDITING

Finally, we address image editing using the FLUX Kontext model (Labs et al., 2025), which, as we find, can struggle with complex edits involving multiple objects. To overcome this, we apply modulation guidance, using the final prompt as the positive direction and a blank prompt as the negative. We validate our approach on the SEED-Data benchmark (Ge et al., 2024) and present the results and implementation details in Appendix F. Representative examples are shown in Figure 8.

## 7 CONCLUSION

In this paper, we revisit the role of the pooled text embedding, showing that, despite its weak influence, it can improve performance across tasks and models when used from a different perspective. We present ablation studies in Appendix C, where dynamic modulation guidance outperforms constant guidance, offering greater flexibility for practitioners. Limitations are discussed in Appendix H.

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

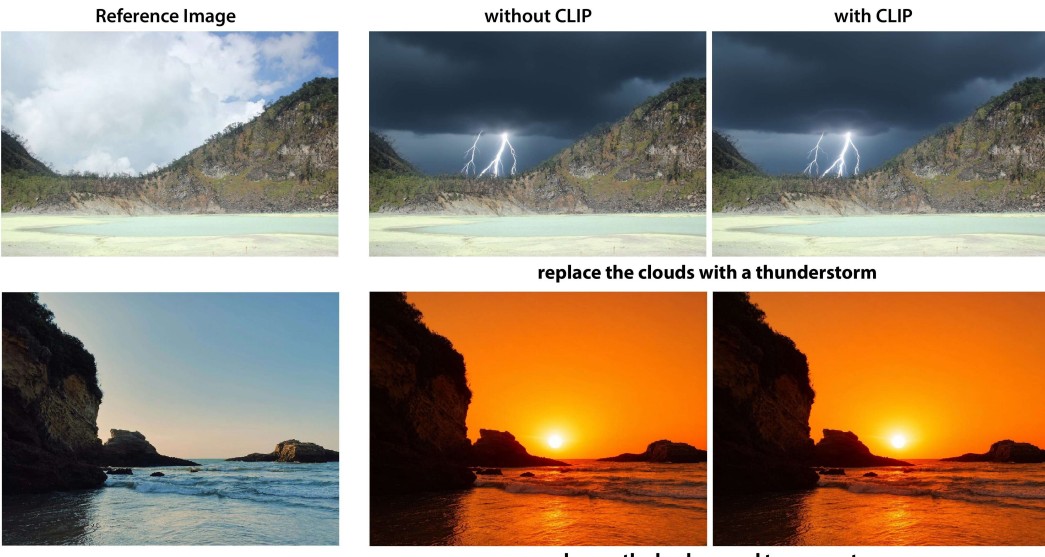

Figure 9: We observe that the CLIP text encoder does not influence instruction-guided image editing performed with the FLUX kontext model.

Table 5: Configuration of hyperparameters for dynamic modulation guidance

| Task | Positive prompt | Negative prompt | Guidance strategy |
|---|---|---|---|
| Text-to-image aesthetics | Ultra-detailed, photorealistic, cinematic | Low-res, flat, cartoonish | Strategy 1 in Figure 3(b) $i = 5$, $w = 3$ |
| Text-to-image complexity | Extremely complex, the highest quality | Very simple, no details at all | Strategy 1 in Figure 3(b) $i = 10$, $w = 3$ |
| Text-to-image hands correction | Natural and realistic hands | Unnatural hands | Strategy 4 in Figure 3(b) $i_1 = 13, i_2 = 30, i_3 = 45$ $w_1 = 3, w_2 = 1$ |
| Text-to-image object counting | $[n]$ [objects] | Very simple, no details at all | Strategy 1 in Figure 3(b) $i = 5$, $w = 3$ |
| Text-to-video | Ultra-detailed, photorealistic, cinematic | Low-res, flat, cartoonish | Strategy 1 in Figure 3(b) $i = 5$, $w = 3$ |
| Image editing | Textual prompt | — | Strategy 1 in Figure 3(b) $i = 5$, $w = 3$ |

# APPENDIX

# A  ADDITIONAL ANALYSIS FOR FLUX KONTEXT MODEL

Here, we analyze the impact of the CLIP model on FLUX Kontext (Labs et al., 2025). We find that dropping the pooled embedding does not affect editing results, as visually confirmed in Figure 9. In addition, we evaluate performance on the SEED-Data benchmark (Ge et al., 2024) with and without the pooled text embedding. We compute the CLIP score (Hessel et al., 2021) to measure reference preservation and prompt correspondence. The results in Table 6 confirm the observation.

Table 6: Editing quality for the FLUX kontext model (with and without CLIP). CLIP has no effect on the model.

| Configuration | CLIP Score, Image ↑ | CLIP Score, Text ↑ |
|---|---|---|
| CLIP+T5 | 79.3 | 29.3 |
| w/o CLIP | 80 (+0.7) | 29.3 (0) |

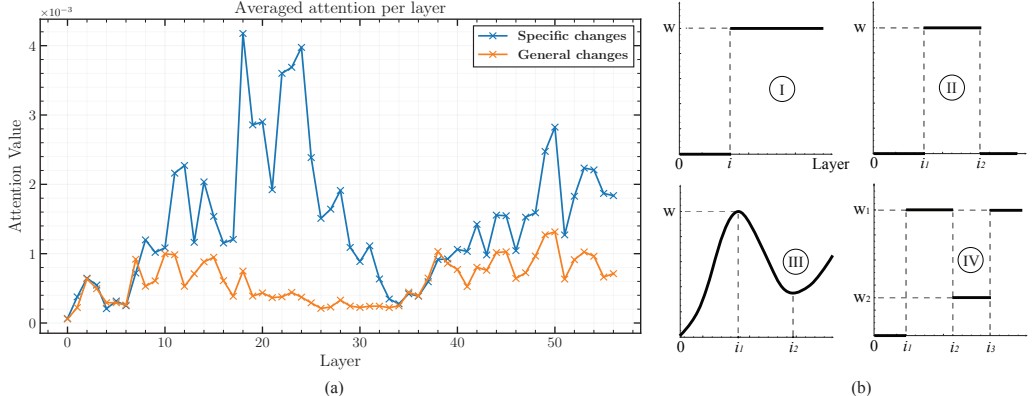

Figure 10: Analysis on dynamic modulation guidance. To derive a dynamic guidance scale, we (a) analyze how the model allocates attention to different features by computing averaged attention maps over two token groups (specific and general). Building on this, we (b) explore dynamic strategies for setting layer-specific $w$ values.

The lack of impact in the editing case may stem from the out-of-distribution nature of instructions for the CLIP model. We find that this mismatch can lead to a lack of editing strength, particularly in complex scenes with multiple objects. To address this, we propose using the final prompt as the CLIP input and applying modulation guidance.

## B  STRATEGIES FOR DYNAMIC GUIDANCE

Recent studies show that attention layers in transformer models specialize at different depths, with each layer focusing on distinct levels of semantic detail (Avrahami et al., 2025). This insight encourages us to investigate which parts of the attention stack are most appropriate for injecting guidance, depending on the desired effect. For example, if fine-grained attributes such as hands are mainly shaped by mid-layer attention, then targeting guidance at those specific layers is more effective and reduces the risk of unintended modifications in other regions of the image.

Thus, we construct two prompt subsets of $1,000$ examples each: one targeting local features (e.g., `hands`, `face`, `eyes`) and the other targeting global features (e.g., `realism`, `cinematic`, `crisp`). We then generate images for each subset and collect the corresponding attention maps for each target aspect. Finally, we average these maps across all examples and present the results for different layers in Figure 10(a). We observe that the model primarily focuses on local features in two layer regions: layers 10–30 and 42–58. In contrast, attention to global features remains relatively constant, with a slight drop between layers 20 and 35.

Based on this analysis, we propose applying dynamic modulation guidance at the layer level. We present four possible strategies in Figure 10(b), with strategies 3 and 4 designed to resemble the observed attention behavior for specific changes. Interestingly, in Appendix C, we find that these strategies provide better results for `hands correction`. For global changes, the step function (case 1) performs well, outperforming the constant scale. Despite introducing additional hyperparameters, our dynamic guidance offers an extra degree of improvement for practitioners, which we believe is important in real-world applications.

## C  ABLATION STUDY

**Dynamic modulation guidance.** First, we ablate different dynamic modulation guidance strategies. Specifically, we consider the FLUX schnell model, testing it on the aesthetics, hands correction, and object counting aspects.

We consider different dynamic guidance strategies from Figure 10(b) and compare them to a constant value of $w = 3$. For dynamic strategies, we use the following parameters.

Table 7: Ablation study of dynamic modulation guidance strategies using human preference (side-by-side win rate). The results demonstrate that dynamic guidance outperforms a constant guidance approach.

| Configuration | | Constant | Strategy 1 | Strategy 2 | Strategy 3 | Strategy 4 |
|---|---|---|---|---|---|---|
| Hands correction | Original | 52 | 48 | 49 | 45 | 41 |
| | Ours | 48 (−4) | 52 (+4) | 51 (+2) | 55 (+10) | 59 (+18) |
| Object counting | Original | 50 | 39 | 40 | 45 | 39 |
| | Ours | 50 (−0) | 61 (+22) | 60 (+20) | 55 (+10) | 61 (+22) |
| Aesthetics | Original | 38 | 28 | 43 | 43 | 46 |
| | Ours | 62 (+24) | 72 (+44) | 57 (+14) | 57 (+14) | 54 (+8) |

**Original, FLUX schnell**  **Constant modulation guidance**  **Dynamic modulation guidance**

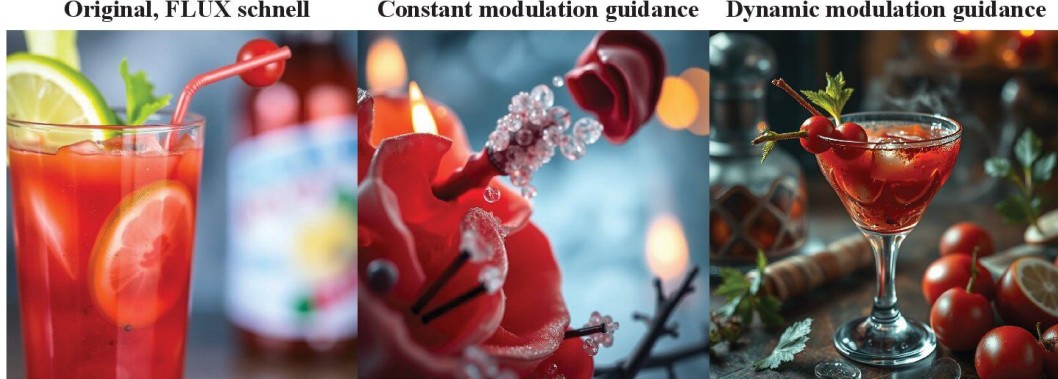

*a close-up of a bloody mary cocktail*

Figure 11: Qualitative comparison of modulation strategies for aesthetics. Constant guidance can overweight the original prompt, leading to significant divergence, whereas dynamic guidance better balances quality and prompt correspondence, allowing the use of larger $w$ without degradation.

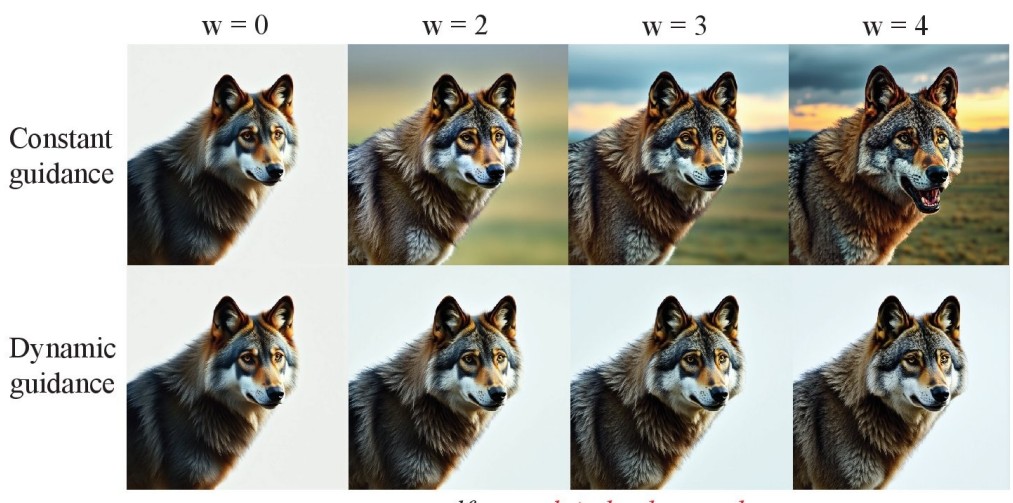

*a wolf on a plain background*

Figure 12: We find that dynamic modulation guidance improves image content (e.g., makes the wolf's fur more detailed) while preserving prompt correspondence. In contrast, constant scales can neglect the prompt request even at small scales (w=2).

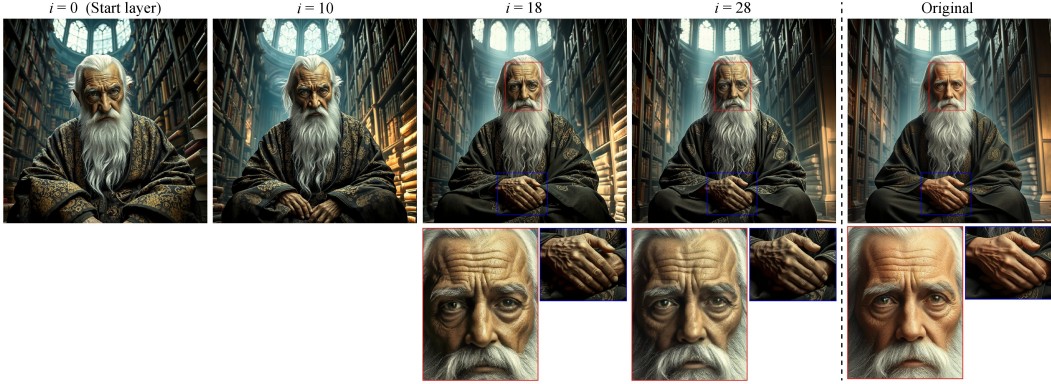

*Imagine a meticulously detailed, hyperrealistic portrait of an aged sage with piercing eyes and a flowing, white beard ...*

Figure 13: Influence of starting layers for complexity guidance. Different choices of $i$ with fixed $w = 3$ illustrate how earlier or later starting layers balance between preserving the original image and improving complexity. In particular, $i = 18$ and $i = 28$ preserve the overall image while enhancing fine-grained details such as faces and hands.

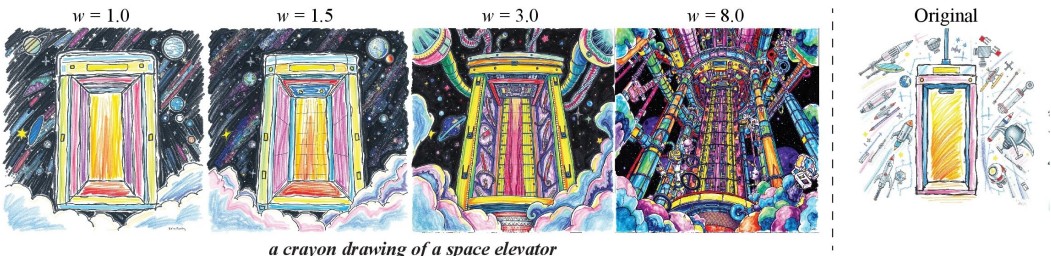

*a crayon drawing of a space elevator*

Figure 14: Influence of guidance strength $w$ for aesthetics. With fixed $i = 5$, increasing $w$ improves image quality by boosting the main object (the elevator) and background details. However, excessively large values, such as $w = 8.0$, can introduce artifacts.

- **Strategy 1.** $i = 5, w = 3$;
- **Strategy 2.** $i_1 = 13, i_2 = 30, w = 3$;
- **Strategy 3.** We use two exponential functions with centers at $i_1 = 20, i_2 = 50$, and $w = 3$;
- **Strategy 4.** $i_1 = 13, i_2 = 30, i_3 = 45, w_1 = 3, w_2 = 1$.

Strategies 3 and 4 are designed to follow the attention pattern illustrated in Figure 10(a).

We conduct a human preference study comparing these strategies to the original model, with results presented in Table 7. First, we observe that dynamic strategies yield higher performance gains compared to a constant scale for hands correction and object counting. Moreover, strategy 4 demonstrates the best performance on hands correction, which aligns with the analysis of attention behavior. For object counting, strategies 1 and 4 perform equally well. We therefore select strategy 1 for this aspect due to its simplicity.

Second, for aesthetics guidance, we observe that strategy 1 achieves the best results, while constant guidance also performs well. However, we find that a constant $w$ can introduce artifacts. As shown in Figure 11, constant guidance can overweight the original prompt, causing significant divergence from the source image. In contrast, dynamic guidance achieves a better balance between quality enhancement and prompt correspondence, enabling the use of higher $w$ values without introducing artifacts as shown in Figure 12.

**Influence of guidance strength and starting layer number.** Next, we analyze how the results change across different starting layers $i$ and modulation guidance strengths $w$. Our main dynamic strategy is the step function (strategy 1 in Figure 3b), and we ablate different choices for this strategy.

FLUX dev, CFG = 1.5          CFG = 2.5          CFG = 3.5

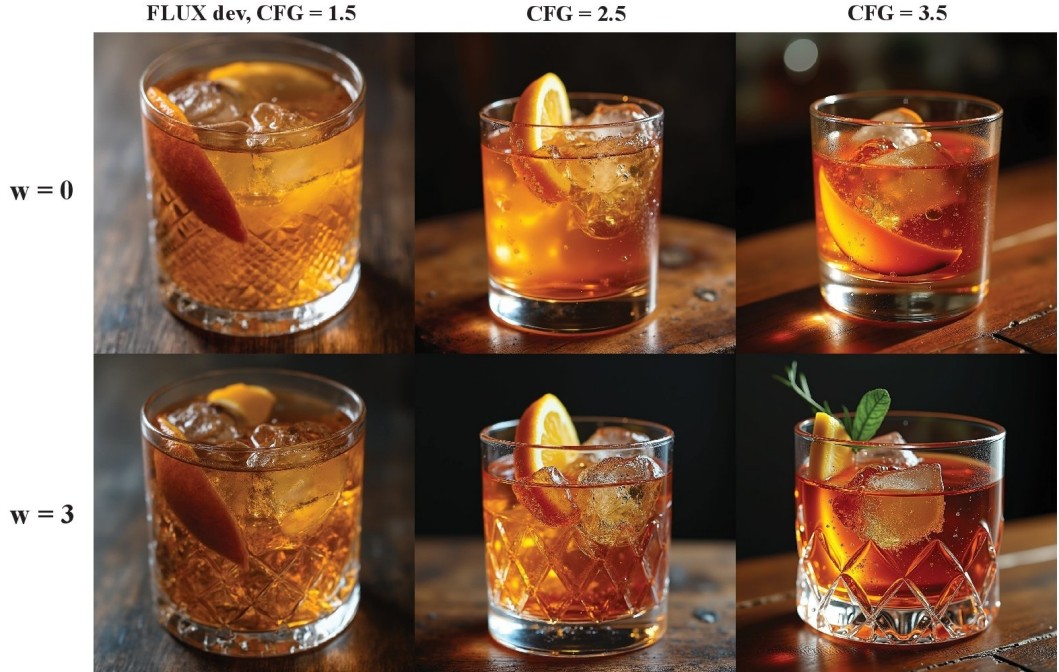

w = 0

w = 3

*a close-up of an old-fashioned cocktail*

Figure 15: We apply modulation guidance across different CFG values and observe consistent improvements, confirming that it is complementary to CFG.

Specifically, in Figure 13, we evaluate different starting layers $i$ with a fixed $w = 3$ under complexity guidance. This setting allows us to balance original image preservation with complexity improvement. In particular, $i = 18$ and $i = 28$ fully preserve the original image while enhancing only fine-grained details such as face and hands.

Then, in Figure 14, we examine the influence of different $w$ values with a fixed starting layer $i = 5$ under aesthetics guidance. We observe that higher $w$ enhances the main object (e.g., the *elevator* in the example) but also improves background details. However, excessively large values, such as $w = 8$, may introduce artifacts.

**Modulation guidance for different CFG.** Finally, we examine how modulation guidance behaves under different CFG values, demonstrating that it can operate effectively on top of CFG. Using the FLUX dev model with complexity guidance, we evaluate multiple CFG values in combination with modulation guidance. The results in Figure 15 show that modulation guidance improves performance across different CFG values, confirming that it is complementary to CFG.

## D    HYPERPARAMETERS CHOICE

In Table 5, we provide the hyperparameters configuration used in our experiments.

For general changes (aesthetics and complexity), we use positive and negative prompts, following the quality-improving prompt modifiers commonly adopted in DMs (Oppenlaender, 2024). In both cases, we employ strategy 1 for dynamic modulation guidance with $w = 3$, but vary the starting layer. Specifically, for complexity, we apply guidance at deeper layers to better preserve the original content while refining high-frequency details.

For specific changes (hands correction and object counting), we adopt strategies 1 and 4, as suggested by the ablation study. For hands correction, we use simple positive and negative prompts: `Natural and realistic hands` and `Unnatural hands`. For object counting, the positive direction

Table 8: Comparison with baselines for **general changes**. We use Normalized Attention Guidance and LLM-enhanced prompts as baselines, and conduct human evaluation on two criteria—**aesthetics** and **complexity**—reporting the corresponding win rates.

| Model | Variant | Aesthetics | | Complexity | |
|---|---|---|---|---|---|
| | | Baseline | Variant | Baseline | Variant |
| *Baseline: LLM-enhanced prompts* | | | | | |
| **FLUX schnell** | Ours | 45 | **55** (+10) | 38 | **62** (+24) |
| **FLUX schnell** | Ours + LLM-enhanced | 39 | **61** (+22) | 26 | **74** (+48) |
| **COSMOS** | Ours + LLM-enhanced | 41 | **59** (+18) | 35 | **65** (+30) |
| *Baseline: Normalized Attention Guidance* | | | | | |
| **FLUX schnell** | Ours | 33 | **67** (+34) | 21 | **79** (+58) |

Table 9: Comparison with baselines for **specific changes**. We use Concept Sliders and LLM-enhanced prompts as baselines, and conduct human evaluation on two criteria: **defects** for hands correction and **text relevance** for object counting, reporting the corresponding win rates.

| Model | Variant | Defects, Hands | | Text relevance, Counting | |
|---|---|---|---|---|---|
| | | Baseline | Variant | Baseline | Variant |
| *Baseline: LLM-enhanced prompts* | | | | | |
| **FLUX schnell** | Ours | 26 | **74** (+48) | 39 | **61** (+22) |
| *Baseline: Concept Sliders* | | | | | |
| **FLUX schnell** | Ours | 42 | **58** (+16) | — | — |

is adapted per prompt but follows a general structure: $[n][\text{objects}]$, where the main object and desired count are taken from the prompt.

For text-to-video generation, we use the same configuration as in aesthetics guidance for text-to-image generation. We find that this not only makes the videos more realistic but also significantly improves their dynamic degree.

For image editing, we adopt the configuration commonly used in CFG: the original prompt serves as the positive direction and a blank prompt as the negative. This setup increases editing strength in cases where the base FLUX Kontext model struggles. For this setting, we use strategy 1.

## E  BASELINES COMPARISONS FOR TEXT-TO-IMAGE GENERATION

We compare our approach against the following baselines: Normalized Attention Guidance (Chen et al., 2025), used for general changes; Concept Sliders (Gandikota et al., 2024), applied to hands correction; and LLM-enhanced prompts (Oppenlaender, 2024), which we consider for both general and specific changes.

For the LLM-enhanced baseline, we use an LLM to modify the prompt sets by adding additional beautifiers, following the same structure used to construct the positive directions in modulation guidance. For the other approaches, we adopt the default configurations provided in their respective papers.

We present the results for **general changes** in Table 8. We observe significant improvements over Normalized Attention Guidance for both criteria (aesthetics and complexity). Importantly, our method does not incur additional overhead, unlike Normalized Attention Guidance, which requires extra passes through computationally intensive attention layers. Second, we find that our approach can be applied on top of LLM-enhanced prompts and brings additional improvements. This is especially important in practice, where different modifiers are commonly applied to basic prompts (Ramesh et al., 2022).

Table 10: Comparison of editing performance measured by VLM scores for **Editing Strength** and **Reference Preservation**.

| Configuration | Editing Strength ↑ | | | | Reference Preservation ↑ | | | |
|---|---|---|---|---|---|---|---|---|
| | Material | Object | Style | Replace object | Material | Object | Style | Replace object |
| Flux Kontext | 66 ±4 | 78 ±2 | 68 ±5 | 71 ±5 | 93 ±0.1 | 92 ±0.3 | 77 ±1 | 90 ±2 |
| Flux Kontext w/o CLIP | 69 (+3) | 78 (0) | 68 (0) | 71 (0) | 93 (0) | 93 (+1) | 79 (+2) | 90 (0) |
| Flux Kontext using final prompt for CLIP | 69 (+3) | 75 (−3) | 68 (0) | 73 (+2) | 93 (0) | 93 (+1) | 80 (+3) | 89 (−1) |
| **Flux Kontext, modulation guidance** | **79** (+13) | **81** (+3) | **72** (+4) | **78** (+7) | 93 (0) | 92 (0) | 78 (+1) | 89 (−1) |

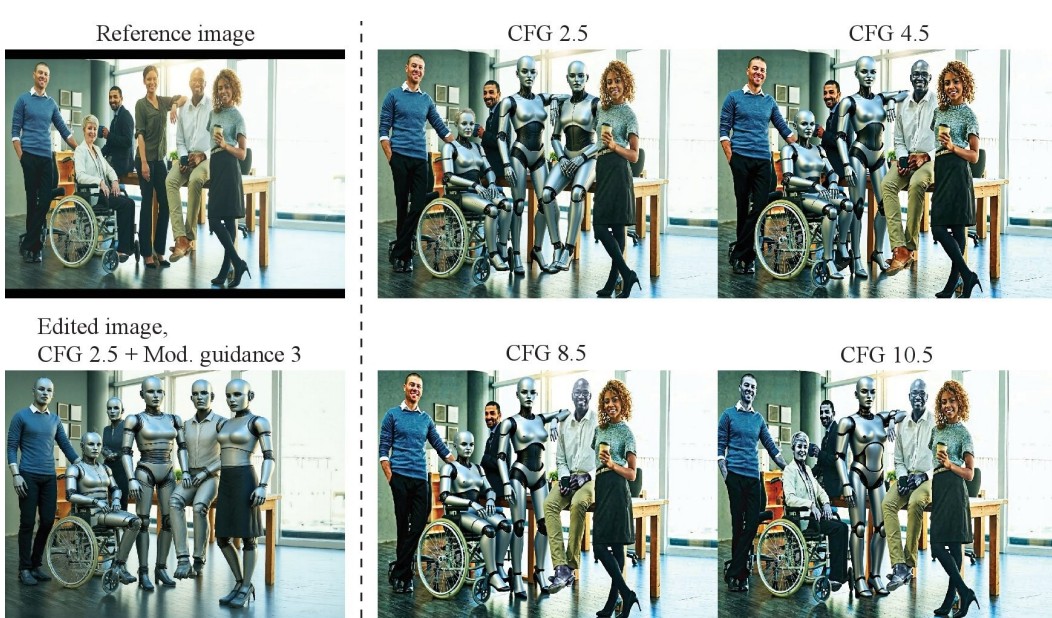

Figure 16: We find that the FLUX Kontext model sometimes struggles with complex image edits, and even higher CFG values do not alleviate this issue. In contrast, modulation guidance can effectively address such cases.

We present the results for **specific changes** in Table 9. First, we find that our approach outperforms the LLM-enhanced prompt baseline on both tasks (hands correction and object counting). Notably, for hands correction, the LLM-enhanced prompt approach can lead to divergence—where the model overemphasizes hands and neglects other parts of the image. In contrast, our approach localizes model attention without adversely affecting the rest of the image. Second, we find that our approach even brings improvements over the Concept Sliders approach, without requiring test-time optimization.

## F    INSTRUCTION-GUIDED IMAGE EDITING

Here, we present the numerical results for instruction-guided image editing using the FLUX Kontext model (Labs et al., 2025). Specifically, we evaluate four settings: (1) the original model; (2) the model without CLIP; (3) the model using the final textual prompt instead of the editing instruction for CLIP; and (4) the model with modulation guidance. For the latter, we use the final prompt as the positive prompt and a blank prompt as the negative, as summarized in Table 5.

To evaluate performance, we follow the basic setting of FLUX Kontext and generate images using the SEED-Data benchmark (Ge et al., 2024), which provides reference images, editing instructions, and final textual prompts. Evaluation is conducted with a VLM model (Bai et al., 2025), which is asked to assess editing strength and reference preservation on a $0 - 100$ scale. For this purpose, we provide the VLM with triples consisting of the reference image, the edited image, and the corresponding instruction.

We report the results in Table 10. First, we observe that removing CLIP does not degrade performance and even yields small improvements, further supporting our intuition that CLIP does not contribute

Table 11: Performance of text-to-image DMs with and without modulation guidance (gray) on Aesthetics and Complexity, evaluated with human preferences and automatic metrics for long and short prompts. Human win rates are reported with respect to the original model; green indicates statistically significant improvement, red a decline. For automatic metrics, **bold** denotes improvement over the original model.

| Model | Side-by-Side Win Rate, % | | | | Automatic Metrics, COCO 5k | | | |
|---|---|---|---|---|---|---|---|---|
| | Relevance ↑ | Aesthetics ↑ | Complexity ↑ | Defects ↑ | PickScore ↑ | CLIP ↑ | IR ↑ | HPSv3 ↑ |
| **FLUX schnell, short prompts** | | | | | 21.6 | 30.1 | 6.2 | 7.8 |
| Ours, Aesthetics guidance | 49 | **64** | **81** | **57** | **21.9** | **30.2** | **7.4** | **8.5** |
| **FLUX schnell, long prompts** | | | | | 21.0 | 33.1 | 10.3 | 10.8 |
| Ours, Aesthetics guidance | 48 | **60** | **73** | 50 | **21.2** | **33.3** | **11.0** | **11.3** |

meaningful gains. Second, we find that using the final prompt instead of the editing instruction for the CLIP model leads to inconsistent outcomes—improving material and replacement criteria while degrading performance on object editing. Finally, we observe that modulation guidance consistently provides improvements across all criteria in terms of editing strength.

Specifically, modulation guidance improves performance on complex editing cases, such as those involving multiple objects. As shown in Figure 16, this problem cannot be solved by simply increasing the CFG scale—only modulation guidance provides improvements.

## G ADDITIONAL EXPERIMENTS

Additionally, we report experimental results for long and short prompts separately to demonstrate that our approach works well with long prompts, whereas basic CLIP tends to influence only short prompts. We conduct a quantitative evaluation using prompts from the MJHQ dataset separated into long and short prompts. We calculate automatic metrics using $1,000$ prompts and conducted a human evaluation using 300 prompts. The results are presented in Table 11. We find that our modulation guidance also has a positive impact on long prompts. For instance, human evaluation shows improvements of $+20\%$ in aesthetics and $+46\%$ in image complexity compared to the original model (FLUX schnell).

## H LIMITATIONS

Our approach also has several limitations. First, it does not address text-to-image correspondence, meaning that it cannot improve how accurately the generated image reflects the input prompt. This limitation is inherent to the modulation guidance design, which focuses on enhancing aesthetic quality, complexity, and other visual attributes rather than semantic alignment. Second, our method introduces a small number of additional hyperparameters that must be tuned to achieve optimal performance. While this tuning process is relatively straightforward, it may add an extra step compared to baseline methods that do not require such configuration.

## I MORE VISUAL RESULTS

We provide additional visual comparisons in Figures 17, 18, 19, 20, 21, 22, and 23.

## J HUMAN EVALUATION

The evaluation is conducted using Side-by-Side (SbS) comparisons, where assessors are presented with two images alongside a textual prompt and asked to choose the preferred one. For each pair, three independent responses are collected, and the final decision is determined through majority voting.

The human evaluation is carried out by professional assessors who are formally hired, compensated with competitive salaries, and fully informed about potential risks. Each assessor undergoes detailed training and testing, including fine-grained instructions for every evaluation aspect, before participating in the main tasks.

In our human preference study, we compare the models across four key criteria: relevance to the textual prompt, presence of defects, image aesthetics, and image complexity. Figures 24, 27, 25, 26 illustrate the interface used for each criterion. Note that the images displayed in the figures are randomly selected for demonstration purposes.

## K    ADDITIONAL DISCUSSION

This work involves human evaluations conducted through side-by-side image comparisons to assess model performance across various criteria (e.g., aesthetics, complexity, and defects). All human studies were performed with informed consent, and participants were compensated fairly for their time. No personally identifiable information was collected, and all data were anonymized prior to analysis. Our research uses publicly available datasets and pre-trained models, adhering to their respective licenses and terms of use. While our method aims to improve the quality and controllability of generative models, we recognize the potential for misuse of generative technologies, including the creation of misleading or harmful content. We encourage responsible use and recommend implementing safeguards in real-world applications.

We note that in this paper a large language model (LLM) was used exclusively for polishing the writing. It was not employed to generate ideas, methods, or contributions.

**Ours, aesthetics**  **Original, FLUX schnell**

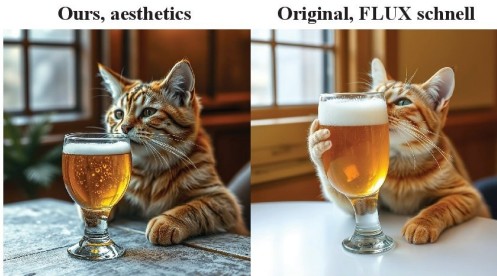 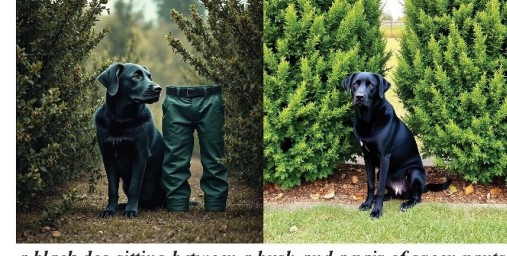

*a cat drinking a pint of beer*

*a black dog sitting between a bush and a pair of green pants standing up with nobody inside them*

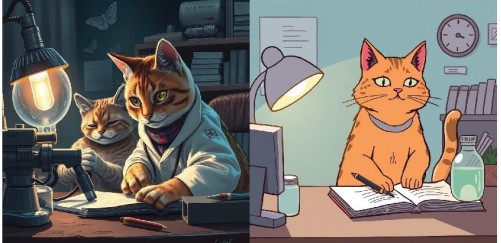 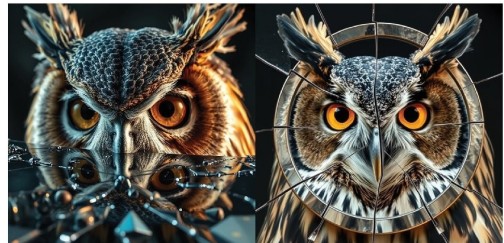

*a comic about two cats doing research*

*long shards of a broken mirror reflecting the eyes of a great horned owl*

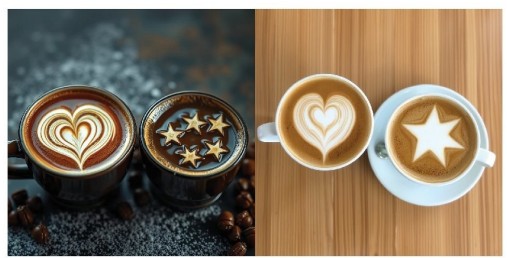 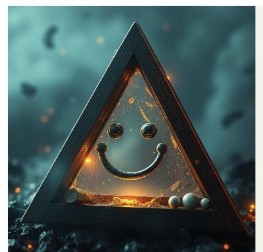 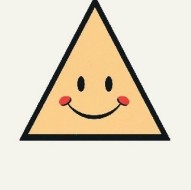

*Two cups of coffee, one with latte art of a heart. The other has latte art of stars.*

*a triangle with a smiling face*

**Ours, complexity**  **Original, FLUX schnell**

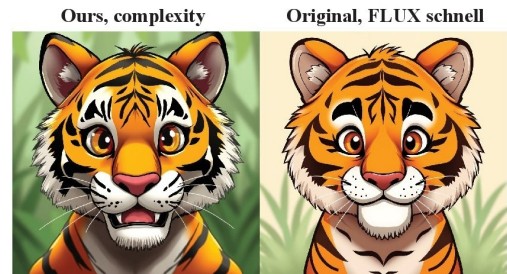 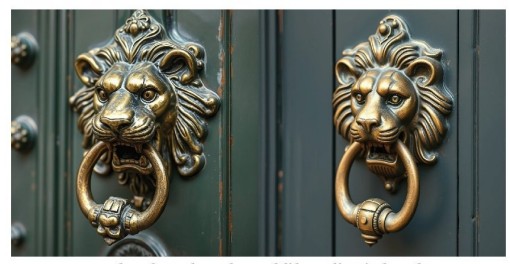

*A cartoon tiger face*

*a doorknocker shaped like a lion's head*

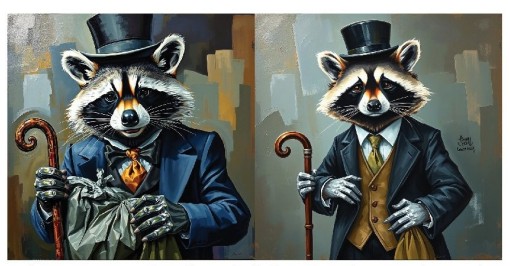 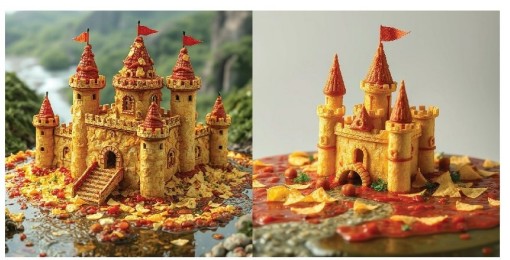

*A raccoon wearing formal clothes, wearing a tophat and holding a cane. The raccoon is holding a garbage bag. Oil painting in the style of abstract cubism.*

*A castle made of tortilla chips, in a river made of salsa. There are tiny burritos walking around the castle*

Figure 17: Visual comparisons for FLUX schnell model

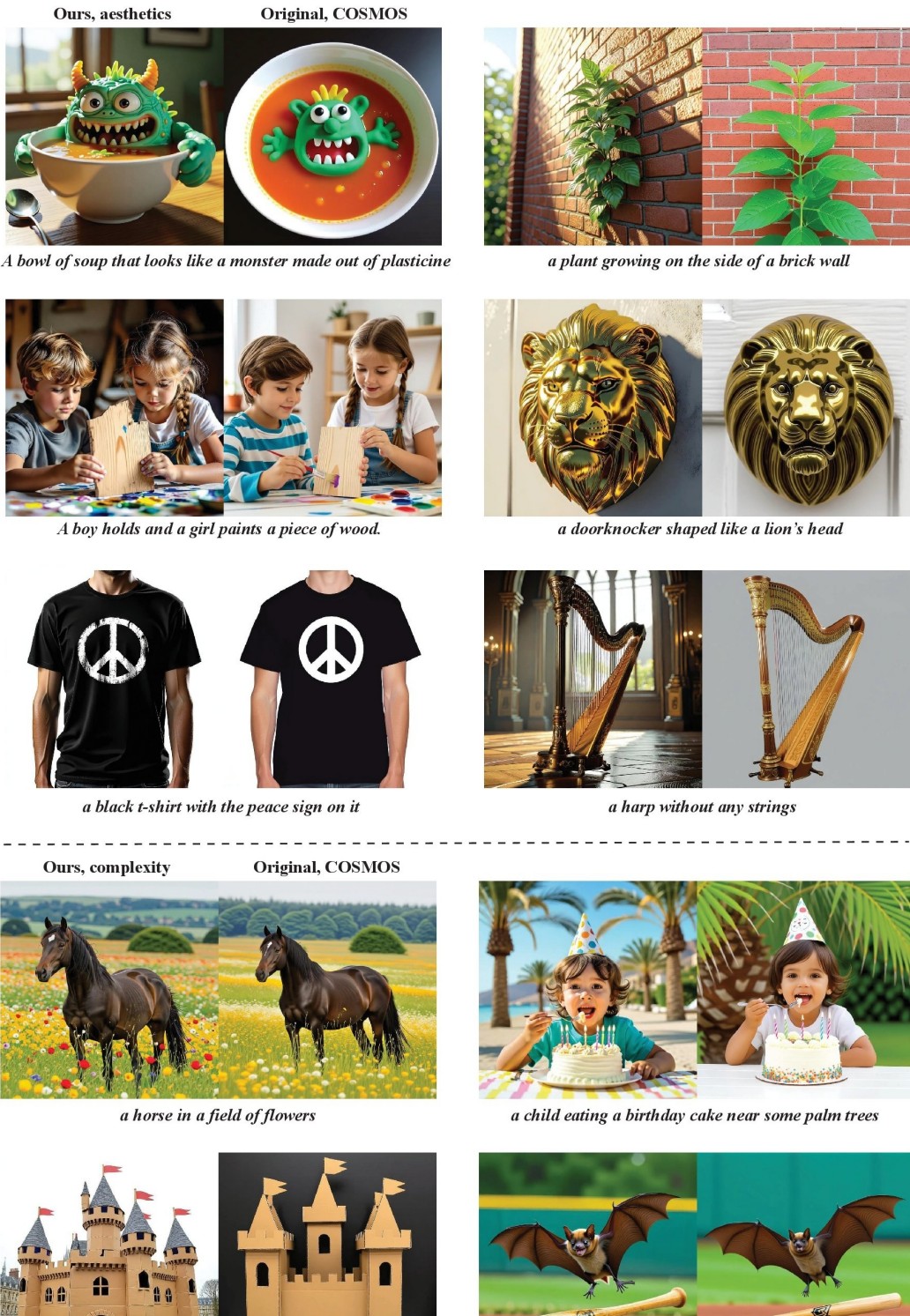

Figure 18: Visual comparisons for COSMOS model

**Ours, aesthetics**  **Original, HiDream**

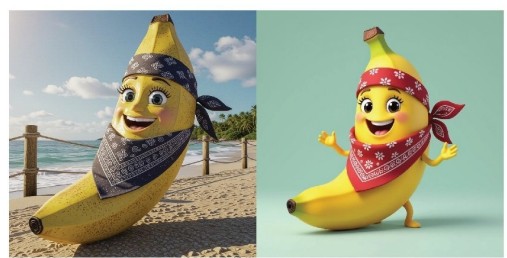

*a small kitchen with a white goat in it*

*a smiling banana wearing a bandana*

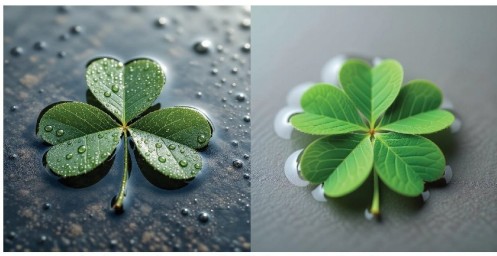

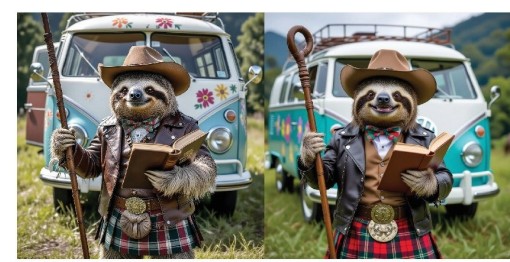

*A photo of a four-leaf clover made of water.*

*A smiling sloth is wearing a leather jacket, a cowboy hat, a kilt and a bowtie. The sloth is holding a quarterstaff and a big book.*

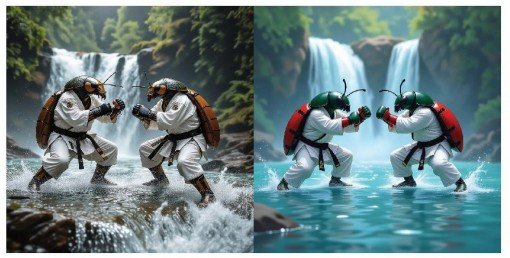

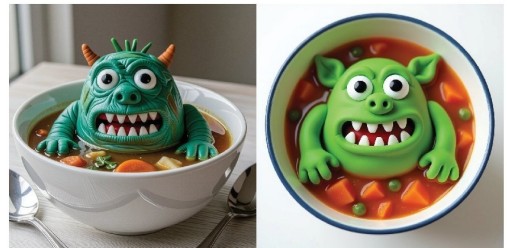

*A close-up of two beetles wearing karate uniforms and fighting, jumping over a waterfall.*

*A bowl of soup that looks like a monster made out of plasticine*

**Ours, complexity**  **Original, HiDream**

*a drawing of a house on a mountain*

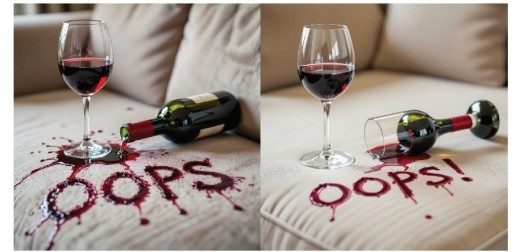

*A glass of red wine tipped over on a couch, with a stain that writes 'OOPS' on the couch.*

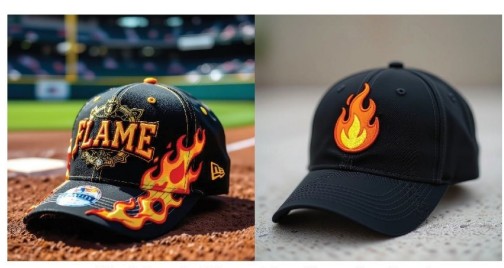

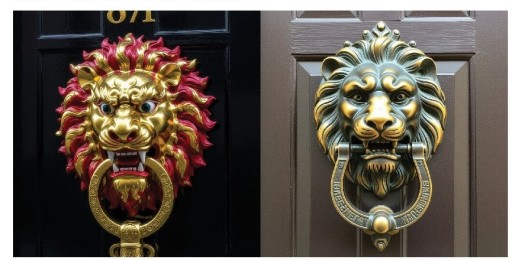

*a black baseball hat with a flame decal on it*

*a doorknocker shaped like a lion's head*

Figure 19: Visual comparisons for HiDream-Fast model

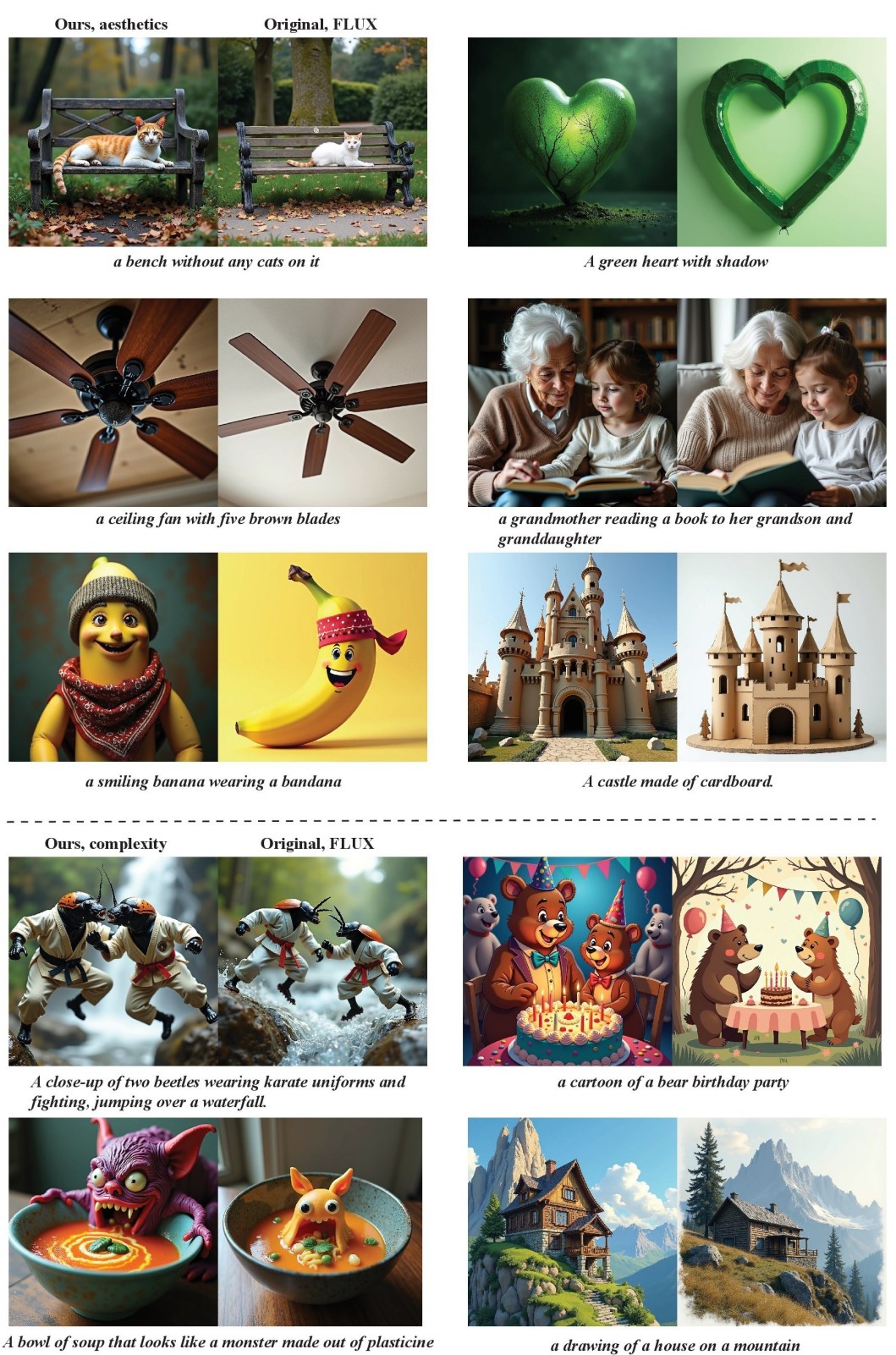

Figure 20: Visual comparisons for FLUX model

**Ours, aesthetics**     **Original, SD3.5 Large**

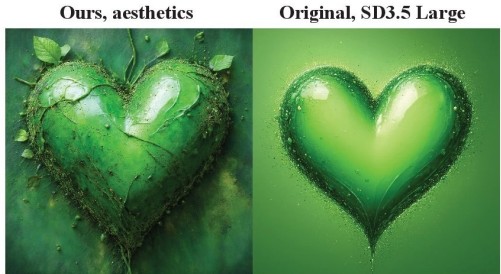 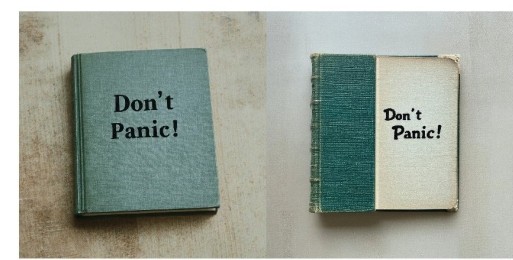

*A green heart*     *a book with the words 'Don't Panic!' written on it*

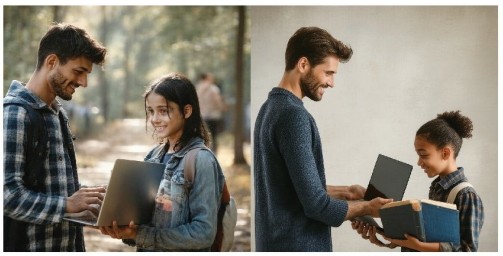 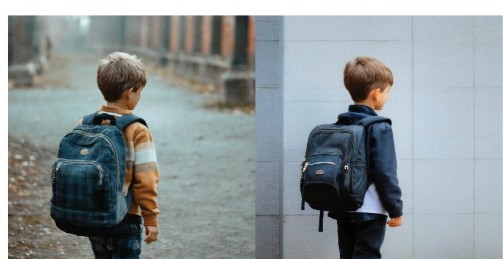

*A man gives a woman a laptop and a boy a book.*     *a boy going to school*

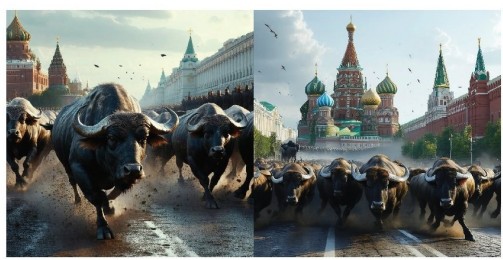 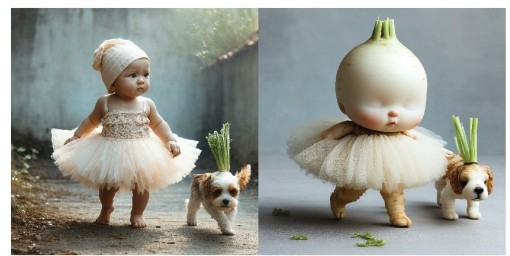

*a herd of buffalo stampeding at the Kremlin*     *a baby daikon radish in a tutu walking a dog*

**Ours, complexity**     **Original, SD3.5 Large**

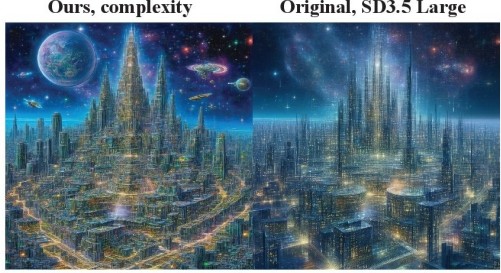 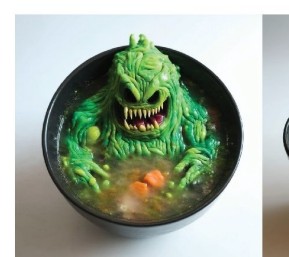 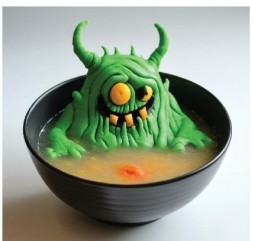

*A city in 4-dimensional space-time*     *A bowl of soup that looks like a monster made out of plasticine*

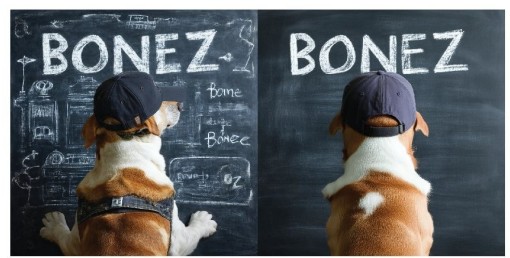 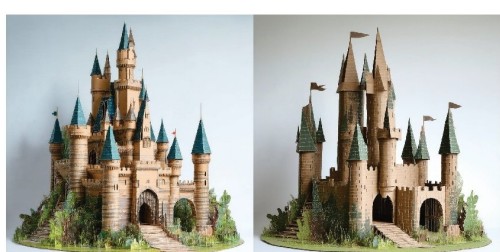

*a dog wearing a baseball cap backwards and writing BONEZ on a chalkboard*     *A castle made of cardboard.*

Figure 21: Visual comparisons for SD3.5 Large model

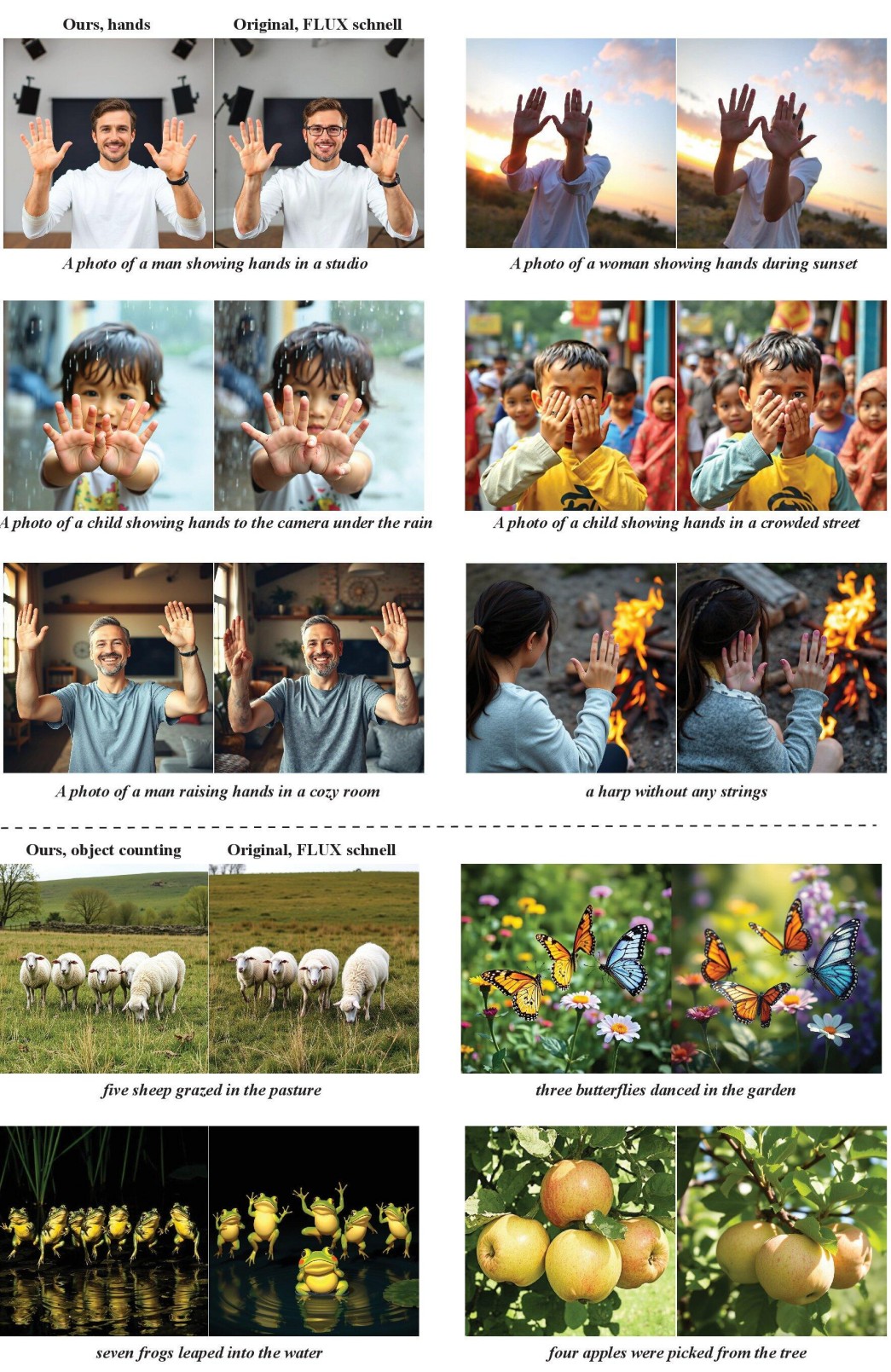

Figure 22: Visual comparisons for FLUX schnell model

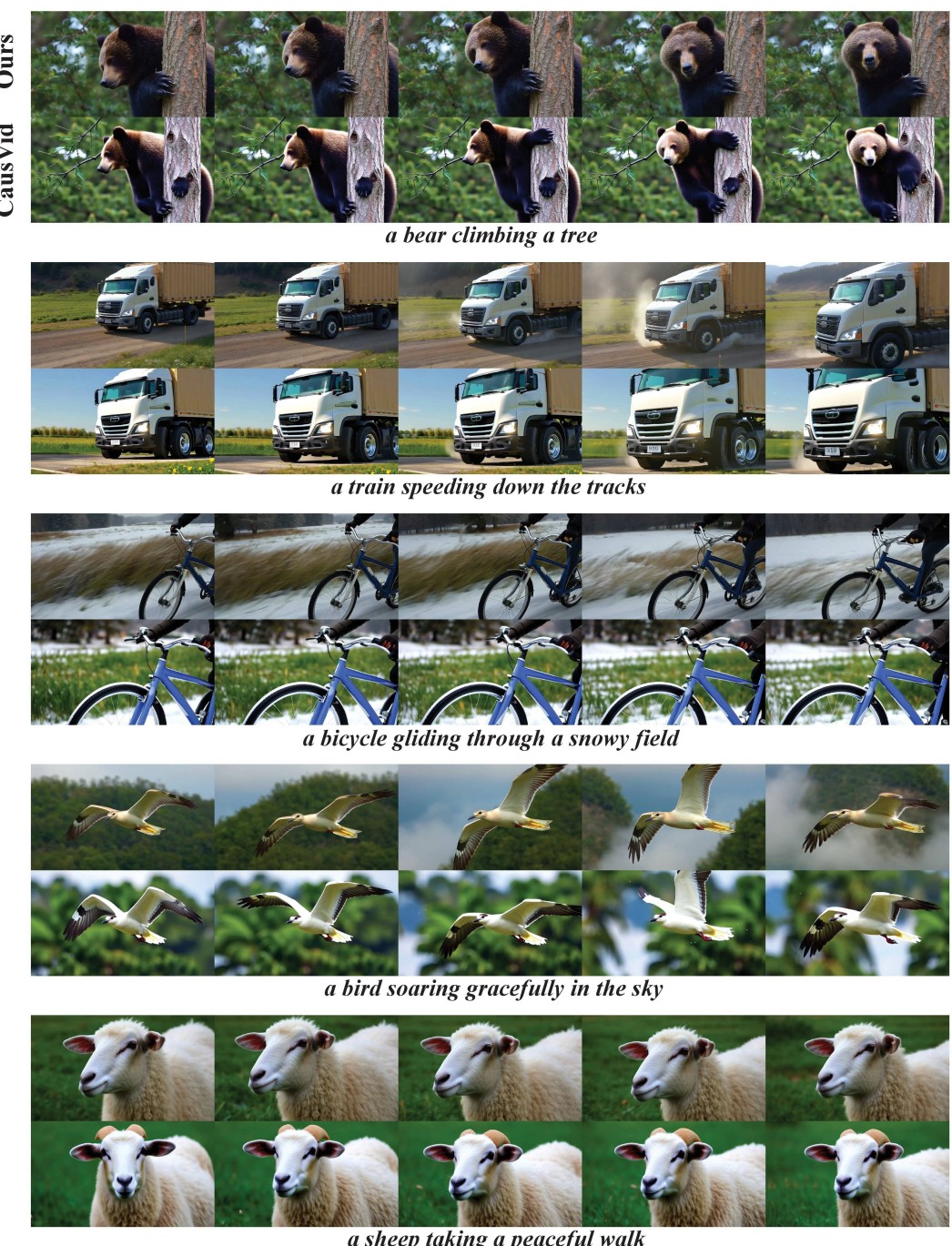

Figure 23: Visual comparisons for CausVid video model

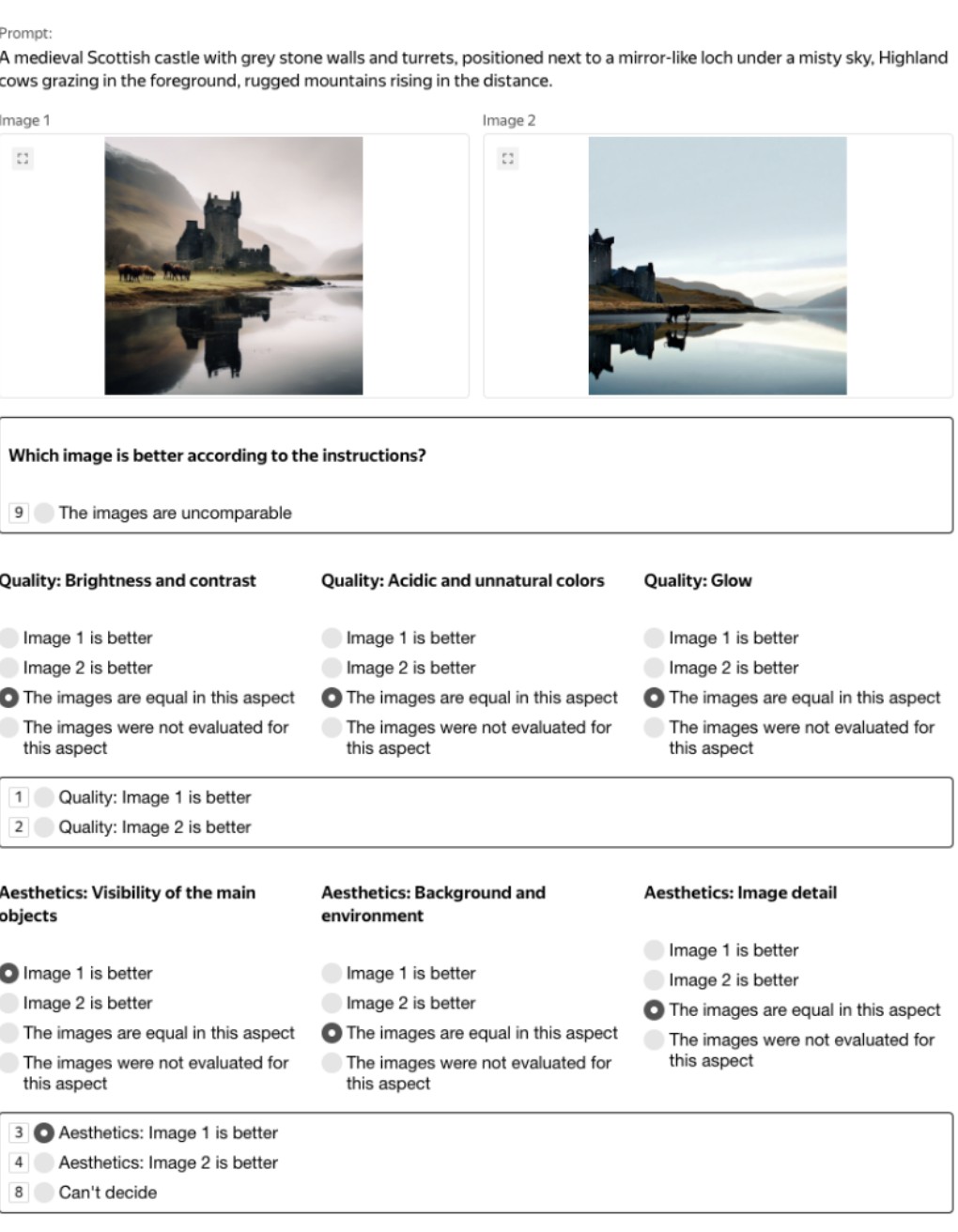

Figure 24: Human evaluation interface for aesthetics.

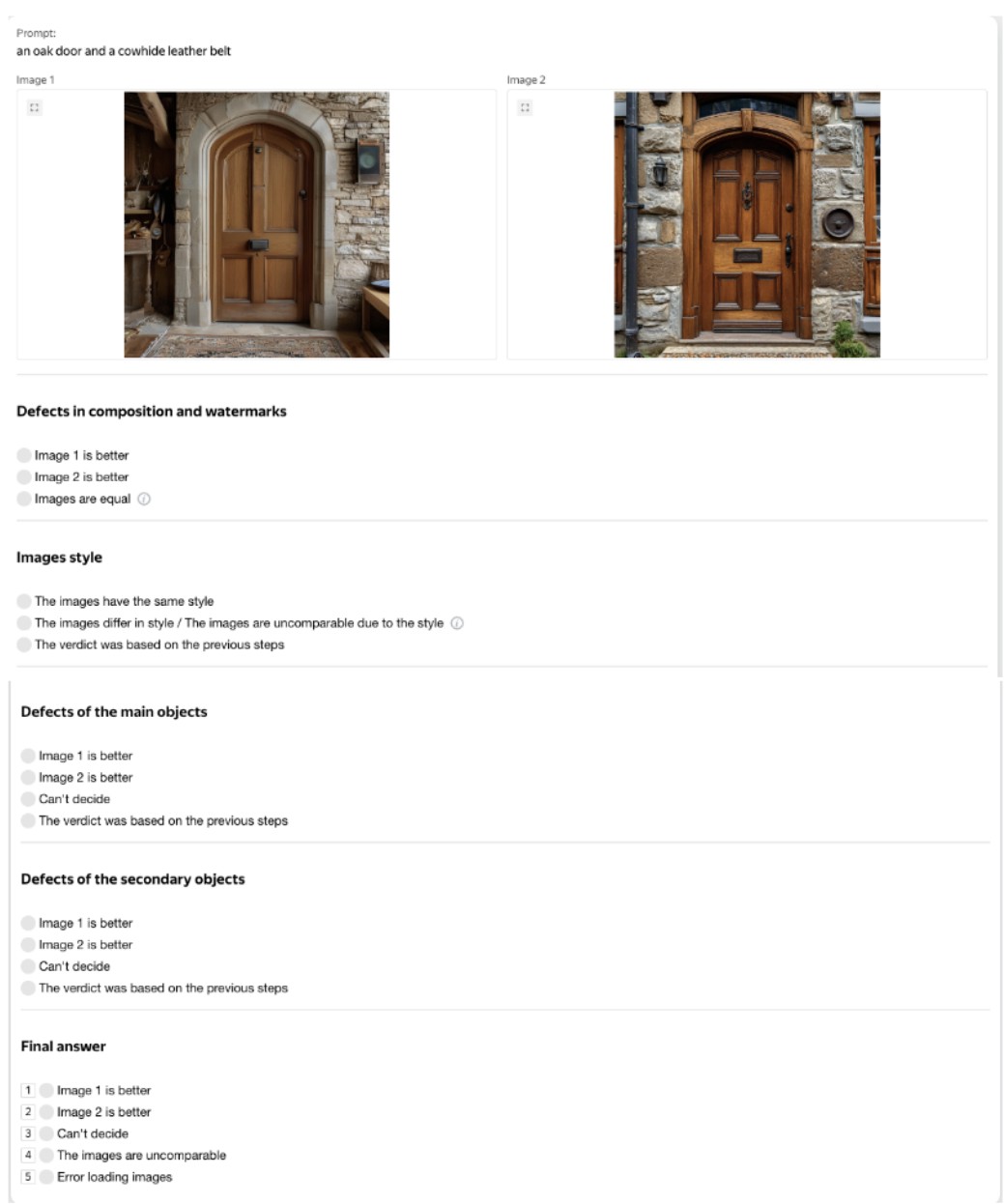

Figure 25: Human evaluation interface for defects.

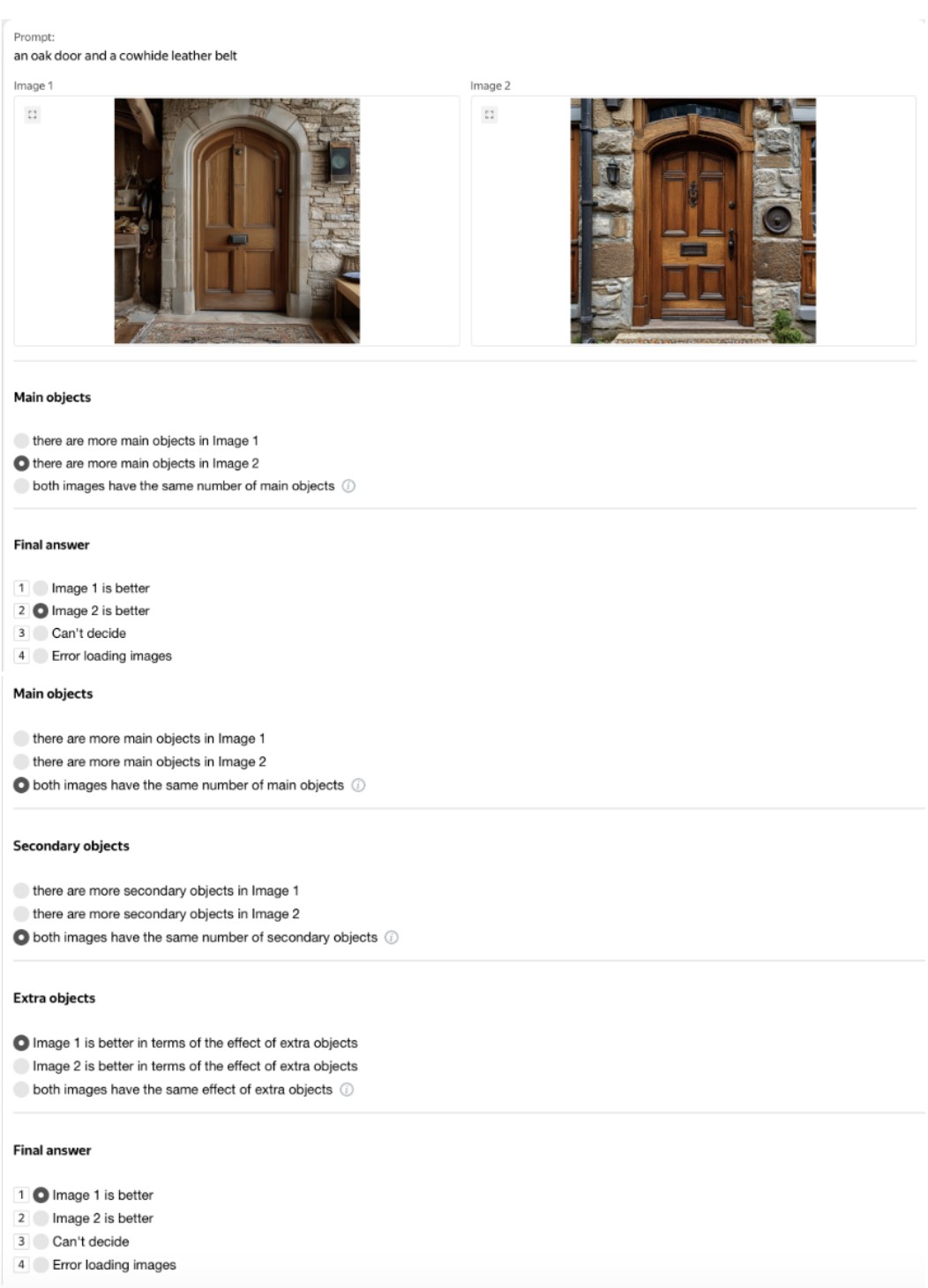

Figure 26: Human evaluation interface for relevance.

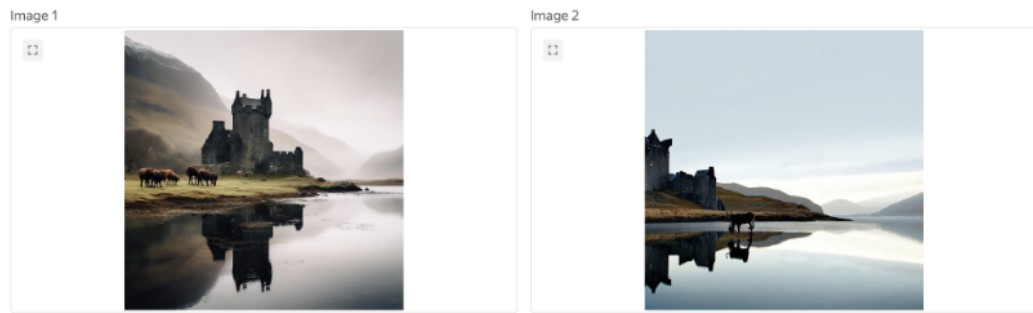

Figure 27: Human evaluation interface for complexity.

