# OpenReview forum: "Rethinking Global Text Conditioning in Diffusion Transformers"
_ICLR.cc/2026/Conference — ICLR 2026 Poster_

### Official Review · Reviewer_DbAo · 2025-10-24

**Soundness:** 3
**Presentation:** 4
**Contribution:** 3
**Rating:** 4
**Confidence:** 4

**Summary:**

This paper explores the impact of modulation-based text conditioning on text-to-image diffusion models. The authors demonstrate that this technique is an important factor in increasing the quality of generated images. They introduce a simple, training-free method that improves performance across various diffusion models without imposing any additional computational burden at inference time.

**Strengths:**

- The paper is well-written, presenting its concepts and results with clarity.
- The proposed approach is easy to apply, computationally inexpensive, and demonstrably improves generation quality.
- The method is validated through a sound evaluation on state-of-the-art models, confirming its effectiveness.
- A significant advantage is the method's broad applicability, as it can be used even with models that do not rely on a CLIP text encoder.

**Weaknesses:**

- Based on the observation presented in Table 1 that adding CLIP embeddings can increase the quality of images generated from short prompts, it would be beneficial if the authors would also separate their evaluation based on the criteria of prompt length, to demonstrate that modulation can increase generation quality even with long prompts. A more detailed analysis would strengthen the method's reliability.
- The evaluation lacks a comparison to common, practical methods for quality enhancement. For example, many users simply add phrases like "good quality image" or "very detailed image" to their prompts or use negative guidance, which is a standard feature in many diffusion model interfaces.
- A potential trade-off between the enforced modulation and prompt fidelity is not explored. If the aesthetic qualities introduced by the modulation contradict the user's explicit request in a prompt, it could negatively impact prompt-following. An exploration of this dynamic would strengthen the submission.

**Questions:**

- The quality improvement shown in Figure 5 appears to stem largely from guiding the outputs to look more photographic, particularly with the introduction of blurred backgrounds. While this is often desirable, it raises a question about the trade-off between this aesthetic guidance and prompt fidelity. For example, if a user explicitly asks for a plain background, will the modulation override this request to produce a more "image-like" result with depth of field? Have the authors evaluated this trade-off?

---

> ### Author Response · Authors · 2025-11-21
> **Reply [1/2]**
>
> We sincerely thank the reviewer for their support and thoughtful questions. The questions raised are extremely important and will certainly help strengthen the submission. We respond to them carefully below.
>
> -------
>
> > ***R1.** Based on the observation presented in Table 1 that adding CLIP embeddings can increase the quality of images generated from short prompts, it would be beneficial if the authors would also separate their evaluation based on the criteria of prompt length, to demonstrate that modulation can increase generation quality even with long prompts. A more detailed analysis would strengthen the method's reliability.*
>
> Thanks for a great suggestion. We have conducted a quantitative evaluation using prompts from the MJHQ dataset separated into long and short prompts (the same setting as in Table 1 in the paper). We calculated automatic metrics using 1,000 prompts and conducted a human evaluation using 300 prompts. We provide the results in the tables below.
>
> * **Automatic metrics for long prompts (1K prompts from MJHQ)**
>
> | Configuration                     | CLIP Score | PickScore    | ImageReward   | HPSv3        |
> |----------------------------------|------------|--------------|---------------|--------------|
> | FLUX schnell                     | 33.1       | 21.0         | 10.3          | 10.8         |
> | FLUX schnell without CLIP        | 32.8 (-0.3)| 21.0 (-0.0)  | 10.4 (+0.1)   | 10.8 (-0.0)  |
> | FLUX schnell + mod. guidance (ours) | 33.3 (+0.2)| **21.2 (+0.2)** | **11.0 (+0.7)**   | **11.3 (+0.5)**  |
>
> * **Automatic metrics for short prompts (1K prompts from MJHQ)**
>
> | Configuration                     | CLIP Score | PickScore    | ImageReward   | HPSv3        |
> |----------------------------------|------------|--------------|---------------|--------------|
> | FLUX schnell                     | 30.1       | 21.6         | 6.2           | 7.8          |
> | FLUX schnell without CLIP        | 29.0 (-1.1)| 21.3 (-0.3)  | 4.5 (-1.7)    | 7.5 (-0.3)   |
> | FLUX schnell + mod. guidance (ours) | 30.2 (+0.1)| **21.9 (+0.3)** | **7.4 (+1.2)**    | **8.5 (+0.7)**   |
>
>
> * **Human evaluation for long prompts (300 prompts from MJHQ)**
>
> | Configuration                     | Text relevance | Aesthetics     | Complexity     | Defects |
> |----------------------------------|----------------|----------------|----------------|---------|
> | FLUX schnell                     |          -      |         -       |         -       |       -  |
> | FLUX schnell without CLIP        | 49             | 49             | 49             | 49      |
> | FLUX schnell + mod. guidance (ours) | 48          | **60 (+20)**       | **73 (+46)**       | 50      |
>
> * **Human evaluation for short prompts (300 prompts from MJHQ)**
>
> | Configuration                     | Text relevance | Aesthetics     | Complexity     | Defects     |
> |----------------------------------|----------------|----------------|----------------|-------------|
> | FLUX schnell                     |       -         |           -     |       -         |       -      |
> | FLUX schnell without CLIP        | 42             | 51             | 51             | 50          |
> | FLUX schnell + mod. guidance (ours) | 49          | **64 (+28)**       | **81 (+62)**       | **57 (+14)**    |
>
> We find that our modulation guidance also has a positive impact on long prompts. For instance, human evaluation shows improvements of +20% in aesthetics and +46% in image complexity compared to the original model (FLUX schnell).
>
> To support this, we present a few qualitative examples for aesthetics modulation guidance using anonymous links:
>
>
> (i) **Three models on long prompts** (FLUX schnell; FLUX schnell without CLIP; FLUX schnell + mod. guidance (ours)): https://imgbox.com/AMslSJmx
>
> We find that images become more complex and of better quality with aesthetics modulation guidance. In contrast, CLIP without the guidance does not affect the images, keeping them practically the same (as stated in the paper).
>
> (ii) **Three models on short prompts:**  https://imgbox.com/1m0eoKib
>
> We observe the same improvements with modulation guidance. As we stated in the paper, in this case, CLIP itself has more influence. For some prompts, the model does not generate an image that corresponds to the prompt without CLIP (e.g., "discopanda"). For others, its influence is again negligible (e.g., "Bunny wearing glasses").

---

> ### Author Response · Authors · 2025-11-21
> **Reply [2/2]**
>
> > ***R2.** The evaluation lacks a comparison to common, practical methods for quality enhancement. For example, many users simply add phrases like "good quality image" or "very detailed image" to their prompts or use negative guidance, which is a standard feature in many diffusion model interfaces.*
>
> Thanks for highlighting a very important baseline. In the paper, we already compared our approach to a common method for quality enhancement and called it LLM-enhanced prompts. We add common beautifiers like "good quality image" to prompts using an LLM. Please see Appendix D (lines 1000-1020) for details.
>
> We find our approach outperforms this one significantly according to human evaluation. To support this, we present qualitative comparisons for two models (FLUX schnell and COSMOS):
>
> https://imgbox.com/ilDg45ZG
>
> https://imgbox.com/xjoIhD2C
>
>
> -------------
>
> > ***R3.** A potential trade-off between the enforced modulation and prompt fidelity is not explored. If the aesthetic qualities introduced by the modulation contradict the user's explicit request in a prompt, it could negatively impact prompt-following. An exploration of this dynamic would strengthen the submission.*
>
> The main **motivation behind dynamic modulation guidance is to improve the trade-off between quality improvement and prompt correspondence** (please see lines 200-204). As we explored in Appendix B (lines 899-904), constant guidance can contradict the user’s explicit request in a prompt (Figure 10). In contrast, dynamic modulation guidance helps to preserve important content while using higher scales w, improving quality more effectively.
>
> We explored this trade-off only qualitatively, not quantitatively. To fill this gap, we calculated image quality (PickScore) and prompt correspondence (CLIP score) for different weight scales for constant and dynamic modulation guidance. We present the results here: https://images2.imgbox.com/fc/a6/IYAfM2J7_o.jpg
>
> It can be clearly seen that dynamic guidance provides a better trade-off than constant guidance. **Our approach improves image quality without compromising prompt correspondence relative to w=0 (the initial model without modulation guidance).**
>
> -------------
>
> > ***R4.** The quality improvement shown in Figure 5 appears to stem largely from guiding the outputs to look more photographic, particularly with the introduction of blurred backgrounds. While this is often desirable, it raises a question about the trade-off between this aesthetic guidance and prompt fidelity. For example, if a user explicitly asks for a plain background, will the modulation override this request to produce a more "image-like" result with depth of field? Have the authors evaluated this trade-off?*
>
> Our dynamic guidance can improve the complexity and aesthetics of images without altering the background, while constant guidance often neglects this. This trade-off is controlled by the starting layer, which we analyzed in Appendix B (lines 908-912). This parameter allows us to balance original image preservation with quality improvement (Figure 11).
>
> We consider two examples that explicitly request a plain background using “complexity” guidance:
> https://imgbox.com/c6jF9UDW
>
> **We find that dynamic modulation guidance improves image content (e.g., makes the wolf's fur more detailed) while preserving prompt correspondence.** In contrast, constant scales can neglect the prompt request even at small scales (w=2).
>
> -------------
>
> We will add all the discussion in the revision. We believe these clarifications address the reviewers' questions, and we are happy to incorporate them into the manuscript or provide any further details as needed.

---

> > ### Author Response · Authors · 2025-11-26
> > **Invitation to discussion**
> >
> > Dear reviewer DbAo, we are looking forward to discussion. In the rebuttal, we:
> > * included the results for long and short prompts separately
> > * motivated dynamic guidance by demonstrating a better trade-off between aesthetic quality and prompt correspondence
> > * included the results regarding the aesthetics–prompt fidelity trade-off
> >
> > We would be happy to discuss any additional details.

---

> > > ### Comment · Reviewer_DbAo · 2025-11-26
> > >
> > > Thanks to the authors for the reply!
> > >
> > > These comments address my concerns, especially the application of dynamic guidance, which looks promising. I appreciate the additional results presented by the authors, as well as the qualitative examples.
> > >
> > > Based on the comments, I raised my score accordingly.

---

### Official Review · Reviewer_rF5b · 2025-10-24

**Soundness:** 4
**Presentation:** 4
**Contribution:** 4
**Rating:** 8
**Confidence:** 3

**Summary:**

This paper investigates the role of the CLIP modulation component within Text-to-Image (T2I) diffusion models. The authors begin by questioning its necessity, demonstrating through ablation that its removal has a minimal impact on overall generation performance. Despite this finding, the authors argue that this component enable controllable shifts in the generation process. They propose a novel method that involves altering the modulation guidance at different blocks of the diffusion transofmer. This claim is supported by a thorough quantitative analysis across several T2I models and is further extended to text-to-video models. The authors show that their method improves generation quality in terms of aesthetics and complexity. They also demonstrate that it mitigates common generation failures, such as incorrect object counts and anomalous finger generation.

**Strengths:**

The paper is clearly written and well-structured. A primary strength lies in its comprehensive experimental validation. The experiments are thorough and are conducted on 4 T2I models that are trained with CLIP modulation, and even included additional model that was not using CLIP, training it to incorporate CLIP modulation. Furthermore, they include text-to-video models, thereby broadening the applicability of their findings.

**Weaknesses:**

The primary weakness is the limited novelty of the method. This method (with the exception of choosing the dynamic modulation strategies) was already presented in [1] as a naive approach (Equation 2). If the authors disagree, I would be happy to discuss and understand the novelty better.

A second, smaller weakness, concerns the justification for the proposed dynamic modulation strategies. These strategies are heuristically derived from observed attention patterns within the model. This reliance is a potential weakness, as attention weights are not always a reliable or faithful indicator [2] of a model's internal semantic processing at different hierarchical levels. While the authors attempt to validate these attention-driven heuristics through an ablation study, the results appear inconclusive and fail to provide a definitive justification for the chosen strategies.

[1] TokenVerse: Versatile Multi-concept Personalization in Token Modulation Space (Garibi and Yadin et al. 2025)
[2] Attention is not Explanation. (Jain et al. 2019)

**Questions:**

- The analysis suggests that CLIP modulation is more influential for short prompts compared to long prompts. Do the authors have a hypothesis for this observed phenomenon?

---

> ### Author Response · Authors · 2025-11-21
> **Reply**
>
> We sincerely thank the reviewer for the positive evaluation and thoughtful feedback. We are delighted that the reviewer found the experiments comprehensive. We address each point below.
>
> -------
>
> > ***R1.** The primary weakness is the limited novelty of the method. This method (with the exception of choosing the dynamic modulation strategies) was already presented in [1] as a naive approach (Equation 2).*
>
> We fully agree that the basic form of the method has been used in previous works, and we highlight this in the paper. However, in our paper, we have two key novelties.
>
> Our **first** novelty lies in the observed phenomenon that **CLIP’s influence is minimal under standard conditioning but becomes substantial when applied through modulation guidance**, which was not previously known. This insight enables significant improvements across different tasks that were previously unattainable.
>
> **Second**, we highlight that modulation guidance has previously been applied only to specific tasks like semantic editing and has not been thoroughly explored as a general tool for quality enhancement. In contrast, **our paper analyzes it in the general case and provides a comprehensive examination**.
>
> Specifically:
>
> (i) we observe that the pooled embedding can substantially influence the generated image, producing both local and global changes (Figure 2);
>
> (ii) we generalize modulation guidance to a dynamic setting, which significantly improves upon basic constant guidance (Table 7);
>
> (iii) we analyze why modulation guidance helps (Figure 4), showing that the model attends more strongly to the desired features after guidance;
>
> (iv) we propose several techniques for the effective incorporation of CLIP embeddings into existing CLIP-free models (lines 264–299)
>
> (v) our examination of the modulation guidance extends to various models (FLUX, SD3.5, COSMOS, WAN) and tasks (image/video generation and editing).
>
> To the best of our knowledge, a comprehensive examination of CLIP's influence on generative performance and the modulation guidance technique has not been presented before.
>
> -------
>
> > ***R2.** Justification for the proposed dynamic modulation strategies. The results appear inconclusive and fail to provide a definitive justification for the chosen strategies.*
>
> The main conclusion behind the proposed dynamic modulation strategies is that **dynamic modulation guidance consistently outperforms constant guidance** (Table 7). And the simplest variant (a step function) already works well in most cases.
>
> We acknowledge that attention-based strategies do not improve results universally. However, they can be useful in certain scenarios (e.g., hand correction), providing practitioners with an additional option to enhance results. We agree that the results may appear inconclusive, and we will therefore reorganize the “Dynamic modulation guidance” section by **moving the attention-based intuition to the supplementary material and present it as a fully optional tool which can potentially improve the performance.**
>
> -------
>
> > ***R3.** The analysis suggests that CLIP modulation is more influential for short prompts compared to long prompts. Do the authors have a hypothesis for this observed phenomenon?*
>
> Thanks for the interesting question. We believe there are two main reasons why CLIP struggles with long prompts.
>
> **First,** **CLIP was trained predominantly on short captions.** Our analysis of LAION-2B caption lengths shows that most captions are short (around 10 tokens), meaning CLIP’s embedding space is naturally optimized for short inputs. For example, a typical caption is: *“a red car parked on a city street.”* We provide our histogram results via an anonymous link: https://images2.imgbox.com/75/20/DJkT34vl_o.jpg
>
> **Second**, CLIP uses a **pooled text embedding, which captures global semantic information.** As a result, detailed or fine-grained information in longer prompts tends to be compressed or ignored, causing the model to focus on a more informative signal from the attention mechanism.
>
> Importantly, we also want to emphasize that CLIP does not provide meaningful signal even for a portion of short prompts, and this subset is non-negligible. We present several examples: https://imgbox.com/wDu4VYV3 The fact that the model produces identical results with and without CLIP suggests that CLIP is not influential.
>
> To summarize, within the FLUX model, **CLIP contributes useful information for certain short prompts**, but for other models (e.g., HiDream), CLIP appears to have negligible impact. Meanwhile, modulation guidance reactivates CLIP in all cases, leading to consistent and significant performance improvements.
>
> -------
>
> We believe these clarifications address the reviewers' questions, and we are happy to incorporate them into the manuscript or provide any further details as needed.

---

> > ### Comment · Reviewer_rF5b · 2025-11-26
> > **Re: Official Comment by Authors**
> >
> > I thank the authors for their detailed response.
> >
> > The clarifications regarding the paper's novelty and the hypothesis concerning CLIP's behavior on short vs. long prompts satisfactorily address my questions.
> >
> > Consequently, I will leave my positive score unchanged.

---

### Official Review · Reviewer_Tihg · 2025-10-25

**Soundness:** 3
**Presentation:** 3
**Contribution:** 3
**Rating:** 6
**Confidence:** 4

**Summary:**

This paper revisits the role of global text conditioning in diffusion transformers. In response to the prevailing trend of abandoning modulation mechanisms in favor of attention-only approaches, the authors demonstrate through analysis that while conventionally used pooled text embeddings contribute limited benefits, repurposing them as a guidance mechanism can effectively adjust the diffusion trajectory toward more desirable attributes. This approach, termed modulation guidance, is training-free, straightforward to implement, and enhances performance across multiple tasks including text-to-image, text-to-video generation, and image editing.

**Strengths:**

1. The revisiting and discovery that global text conditioning can be leveraged as a powerful control signal—rather than being merely a passive input—is novel. The proposed dynamic modulation guidance demonstrates a clear ability to address classic and stubborn challenges in T2I generation, such as hand synthesis and object counting, which is a significant finding.

2. The paper is impressive in its extensive experimental scope, demonstrating effectiveness across a diverse set of tasks—including text-to-image, text-to-video, and instruction-guided editing—and model architectures, encompassing transformer-based DMs and the CLIP-free COSMOS model.

3. The paper is well-structured and easy to follow.

**Weaknesses:**

I have the following two major questions:

1. I noticed that different hyperparameters are used for different tasks and generation types/styles In Tab.5. Could the authors provide more detailed guidance on the process of selecting the appropriate strategy and its associated hyperparameters for a **new, unseen task**? Is this process largely heuristic, requiring manual search for each new situation, or are there general principles or a methodology that can be derived from the observations in Figure 3 to make this selection more systematic?

2. **Connection and Distinction to h-space Methods:** The work [1] demonstrates that diffusion models possess a semantic latent space (h-space) and that rescaling the difference in latent features ($\Delta h$) can control attribute strength. Could the authors discuss the primary distinction between their method and this prior work? Specifically, is it possible to achieve a similar guidance effect by simply rescaling $\Delta h$, analogous to Equation 3 in this paper, instead of explicitly using the CLIP embedding to compute $y(p^+, t) - y(p^-, t)$?

[1] Mingi Kwon, Jaeseok Jeong, Youngjung Uh. "Diffusion Models Already Have a Semantic Latent Space". *ICLR*, 2023.

**Questions:**

Please see my **Weaknesses** part.

---

> ### Author Response · Authors · 2025-11-21
> **Reply**
>
> We sincerely thank the reviewer for the positive evaluation and thoughtful feedback. We are delighted that the reviewer found our findings novel and significant. We address each point below.
>
> --------
>
> > ***R1.** I noticed that different hyperparameters are used for different tasks and generation types/styles In Tab.5. Could the authors provide more detailed guidance on the process of selecting the appropriate strategy and its associated hyperparameters for a new, unseen task? Is this process largely heuristic, requiring manual search for each new situation, or are there general principles or a methodology that can be derived from the observations in Figure 3 to make this selection more systematic?*
>
> Thank you for raising this important point. Our recommendation is straightforward:
>
> **(i)** For a new, unseen task, **we suggest using dynamic modulation guidance with strategy 1 (step function)**, setting **w = 3** (guidance strength) and **i = 5** (start layer). We use this configuration across most tasks and models (according to Tab. 5) and find that it works well without any additional hyperparameter search. Therefore, **we believe that for a new task, this same setting will also work well without any tuning.**
>
> **(ii, Optional)** Then, we suggest first tuning the starting layer **i** while keeping **w** fixed. A larger starting layer allows the user to preserve the initial image more effectively, as discussed in Appendix B, lines 905–912. Furthermore, it is possible to additionally tune **w** with **i** fixed; larger values increase the desired effect, as we discuss in Appendix B, lines 913–916.
>
> **(iii, Optional)** Finally, if a user wants to further improve performance, they may try different strategies shown in Figure 3. This is optional – these strategies are not required for the guidance to function, but they can refine the results. For instance, we find that it can be useful for local changes such as hand correction.
>
> We will clarify this workflow in the revision.
>
> --------
>
> > ***R2.** Connection and Distinction to h-space Methods.*
>
> Thanks for pointing us to this relevant paper. A primary difference is that the h-space methods operate on unconditional diffusion models, whereas our work focuses on the text-conditional case.
>
> This leads to the **first distinction.** To discover semantic directions, methods that operate in the h-space require additional training. In contrast, **our method is fully training-free; it requires no fine-tuning or optimization to discover semantic directions.** This is because our approach is designed for the text-conditional case, where the text conditioning naturally provides the ability to find semantic directions.
>
> The **second distinction is where the guidance acts.** The text-conditioning enables effective mechanisms for image manipulation through cross-attention and modulation spaces [1]. In the unconditional case, such mechanisms are not available, necessitating the use of h-space to guide the model within this space.
>
> *[1] TokenVerse: Versatile Multi-concept Personalization in Token Modulation Space*
>
> We will include this work in the revision.
>
> --------
>
> We believe these clarifications address the reviewers' questions, and we are happy to incorporate them into the manuscript or provide any further details as needed.

---

> > ### Comment · Reviewer_Tihg · 2025-11-26
> > **Official Comment by Reviewer Tihg**
> >
> > Thank the authors for the reply!
> >
> > The proposed method can provide a general parameters setting for new tasks, which address my concern.

---

### Official Review · Reviewer_Jrir · 2025-10-30

**Soundness:** 2
**Presentation:** 2
**Contribution:** 1
**Rating:** 4
**Confidence:** 3

**Summary:**

This paper revisits the role of global (pooled) text conditioning in diffusion transformers, which has recently been discarded in favor of attention-only conditioning. The authors first demonstrate through empirical analysis that the pooled CLIP embedding contributes little to generation quality in several state-of-the-art models (e.g., FLUX, HiDream-Fast), especially with long prompts. However, they propose a novel perspective: repurposing the pooled embedding for modulation guidance—a training-free, plug-and-play technique that steers the diffusion process toward desirable visual properties (e.g., aesthetics, complexity, hand realism) by extrapolating in the modulation space using positive/negative prompt pairs. The method is simple, incurs negligible overhead, works with or without classifier-free guidance (CFG), and can be retrofitted into models that originally lack pooled embeddings. Extensive experiments across text-to-image, text-to-video, and image editing tasks show consistent improvements in human evaluations and automatic metrics.

**Strengths:**

1. The paper provides a clear and convincing empirical investigation into why global text conditioning appears ineffective in current models, filling an important gap in understanding.

2. Modulation guidance is training-free, easy to implement, computationally lightweight, and broadly applicable across architectures and tasks.

**Weaknesses:**

1. The core idea of using pooled embeddings for guidance resembles prior work on semantic directions in GANs (e.g., StyleGAN) and recent methods like TokenVerse or Concept Sliders, though the application to modulation space in diffusion transformers is new.

2. The method introduces new hyperparameters (guidance scale w, layer indices), requiring tuning for different tasks—though ablations help, this adds complexity compared to plug-and-play baselines.

**Questions:**

NA

---

> ### Author Response · Authors · 2025-11-21
> **Reply**
>
> We sincerely thank the reviewer for the important concerns raised. We address each of them carefully below.
>
> --------
>
> > ***R1.** The core idea of using pooled embeddings for guidance resembles prior work on semantic directions in GANs (e.g., StyleGAN) and recent methods like TokenVerse or Concept Sliders, though the application to modulation space in diffusion transformers is new.*
>
> We would like to highlight that semantic directions constitute an entire field that has been explored extensively in both GANs (e.g., [1], [2], [3], [4]) and diffusion models (e.g., [5], [6], [7]). However, despite the maturity of this research area, our paper introduces key novelties by identifying a new form of semantic guidance.
>
> Our **first** novelty lies in the observed phenomenon that **CLIP’s influence is minimal under standard conditioning but becomes substantial when applied through modulation guidance**, which was not previously known. This insight enables significant improvements across different tasks that were previously unattainable with standard CLIP-based conditioning.
>
> **Second**, we highlight that modulation guidance has previously been applied only to specific tasks like semantic editing and has not been thoroughly explored as a general tool for quality enhancement. In contrast, **our paper analyzes it in the general case and provides a comprehensive examination**.
>
> Specifically:
>
> (i) we observe that the pooled embedding can substantially influence the generated image, producing both local and global changes (Figure 2);
>
> (ii) we generalize modulation guidance to a dynamic setting, which significantly improves upon basic constant guidance (Table 7);
>
> (iii) we analyze why modulation guidance helps (Figure 4), showing that the model attends more strongly to the desired features after guidance;
>
> (iv) we propose several techniques for the effective incorporation of CLIP embeddings into existing CLIP-free models (lines 264–299)
>
> (v) our examination of the modulation guidance extends to various diffusion models (FLUX, SD3.5, COSMOS, WAN), few-step models (FLUX schnell, Hi-Dream Fast) and tasks (image/video generation and editing).
>
> To the best of our knowledge, a comprehensive examination of CLIP's influence on generative performance and the modulation guidance technique has not been presented before.
>
> [1] InterFaceGAN - Interpreting the Latent Space of GANs for Semantic Face Editing
>
> [2] GANSpace: Discovering Interpretable GAN Controls
>
> [3] StyleSpace Analysis: Disentangled Controls for StyleGAN Image Generation
>
> [4] StyleCLIP: Text-Driven Manipulation of StyleGAN Imagery
>
> [5] TokenVerse: Versatile Multi-concept Personalization in Token Modulation Space
>
> [6] Diffusion Models already have a Semantic Latent Space
>
> [7] DiffusionCLIP: Text-Guided Diffusion Models for Robust Image Manipulation
>
> --------
>
> > ***R2.** The method introduces new hyperparameters (guidance scale w, layer indices), requiring tuning for different tasks—though ablations help, this adds complexity compared to plug-and-play baselines.*
>
> We appreciate this practical concern. **First**, we would like to highlight that **for most cases we use the same setting with fixed hyperparameters**, and this setting generalizes well across different models and tasks (image/video generation and editing; please see Table 5). Therefore, we believe that for a new task, this same setting will also work well without any tuning.
>
> **Second**, to the best of our knowledge, **there are no plug-and-play baselines similar to our approach that do not require any hyperparameter selection**. Other methods either require fine-tuning (like Concept Sliders) or hyperparameter selection (like Normalized Attention Guidance).
>
> --------
>
> We believe these clarifications address the reviewers' questions, and we are happy to incorporate them into the manuscript or provide any further details as needed.

---

> > ### Author Response · Authors · 2025-11-26
> > **Invitation to discussion**
> >
> > Dear reviewer Jrir, we are looking forward to discussion. Given the limited discussion period, we would greatly appreciate any feedback on our responses so that we can further clarify and address the remaining questions in time.

---

### Author Response · Authors · 2025-11-24
**Global reply with revised paper**

We thank the reviewers for their thorough and diligent work. Their insightful comments and questions have led us to significantly improve the paper.

------

# Revision

**We have uploaded a revised version of the paper and highlighted the changes in blue.** Specifically,

**(i)** For the reviewers $\textcolor[RGB]{179,0,0}{\textbf{Jrir, Tihg, rF5b, DbAo}}$: **we have substantially improved the “Dynamic Modulation Guidance” section** (please see lines 208–243). **First,** we simplified the presentation by focusing on the basic form of dynamic modulation guidance and moved the attention-based discussion to the Appendix **(addressing the concern raised by $\textcolor[RGB]{179,0,0}{\textbf{rF5b}}$)**. **Second,** we clearly motivate dynamic guidance by demonstrating a better trade-off between aesthetic quality and prompt correspondence **(addressing the concern raised by $\textcolor[RGB]{179,0,0}{\textbf{DbAo}}$)**. **Third,** we highlight that the simplest case generalizes well across tasks, suggesting that it can potentially be applied to an unseen task without any additional tuning **(addressing the concern raised by $\textcolor[RGB]{179,0,0}{\textbf{Tihg}}$)**.

**(ii)** For the reviewer $\textcolor[RGB]{179,0,0}{\textbf{DbAo}}$: **we have included the results regarding the aesthetics–prompt fidelity trade-off.** Please see Figure 3(a), Figure 12, and lines 964–968.

**(iii)** For the reviewer $\textcolor[RGB]{179,0,0}{\textbf{DbAo}}$: **we have included the results for long and short prompts separately.** Please see Table 11 and lines 1156–1163.

**(iv)** For the reviewer $\textcolor[RGB]{179,0,0}{\textbf{Tihg}}$: **we have added the h-space work to our related work section.**

------

# Positive things

We appreciate the reviewers’ positive feedback on various aspects, including the interesting findings, extensive experiments, meaningful methodology, and clear writing. We summarize these points below.

**(i). Findings:**
* Reviewer $\textcolor[RGB]{179,0,0}{\textbf{Jrir}}$: The paper provides a clear and convincing empirical **investigation into why global text conditioning appears ineffective in current models, $\textcolor[RGB]{0,120,120}{\textbf{filling an important gap in understanding}}$.**
* Reviewer $\textcolor[RGB]{179,0,0}{\textbf{Tihg}}$: The revisiting and discovery that **global text conditioning can be leveraged as a powerful control signal—rather than being merely a passive input—$\textcolor[RGB]{0,120,120}{\textbf{is novel}}$.**

**(ii). Method:**
* Reviewer $\textcolor[RGB]{179,0,0}{\textbf{Jrir}}$: The method is simple, incurs negligible overhead, and works with models that lack pooled embeddings. Modulation guidance is training-free, easy to implement, computationally lightweight, and broadly applicable across architectures and tasks.
* Reviewer $\textcolor[RGB]{179,0,0}{\textbf{Tihg}}$: The proposed dynamic modulation guidance demonstrates a clear ability to **address classic and stubborn challenges, $\textcolor[RGB]{0,120,120}{\textbf{which is a significant finding}}$.**
* Reviewer $\textcolor[RGB]{179,0,0}{\textbf{rF5b}}$: **$\textcolor[RGB]{0,120,120}{\textbf{A novel method}}$** that involves altering the modulation guidance at different blocks of the diffusion transformer.
* Reviewer $\textcolor[RGB]{179,0,0}{\textbf{DbAo}}$: The proposed approach is easy to apply, computationally inexpensive, and demonstrably improves generation quality. **A significant advantage is the method’s broad applicability.**

**(iii). Experiments:**
* Reviewer $\textcolor[RGB]{179,0,0}{\textbf{Jrir}}$: Extensive experiments show consistent improvements.
* Reviewer $\textcolor[RGB]{179,0,0}{\textbf{Tihg}}$: **The paper is impressive in its extensive experimental scope**, demonstrating effectiveness across a diverse set of tasks.
* Reviewer $\textcolor[RGB]{179,0,0}{\textbf{rF5b}}$: A primary strength lies in its comprehensive experimental validation: four T2I models with CLIP; an additional model that was not using CLIP; text-to-video models.
* Reviewer $\textcolor[RGB]{179,0,0}{\textbf{DbAo}}$: The method is validated through a sound evaluation on state-of-the-art models, confirming its effectiveness.

**(iv). Paper writing:**
* Reviewer $\textcolor[RGB]{179,0,0}{\textbf{Tihg}}$: The paper is well-structured and easy to follow.
* Reviewer $\textcolor[RGB]{179,0,0}{\textbf{rF5b}}$: The paper is clearly written and well-structured.
* Reviewer $\textcolor[RGB]{179,0,0}{\textbf{DbAo}}$: The paper is well-written, presenting its concepts and results with clarity.


------

We would be happy to discuss and include any additional details!

---

### Author Response · Authors · 2025-12-03
**Final reply by the authors**

Dear all,

We thank the reviewers and Area Chairs for their work. We regret that the leakage occurred. To clarify our rebuttal, we provide a brief summary below.

During the discussion, we addressed the concerns raised by the reviewers, and three out of four reviewers have confirmed that our responses satisfactorily addressed their points.

The only reviewer who has not provided confirmation raised concerns regarding (i) the similarity of our approach to prior work and (ii) hyperparameter selection. We would like to note that similar concerns were also raised by Reviewer rF5b (concern i) and Reviewer Tihg (concern ii), both of whom have since confirmed that these concerns have been addressed.

Sincerely, Authors

---

### Meta-Review · Area_Chair_jMuj · 2025-12-26

**Summary:**

This paper explores the use of text conditioning in text-conditioned diffusion models. The authors introduce a training-free approach that improves performance across various models without additional computation. The paper received mixed ratings, with two positive and two negative reviews. One reviewer who gave a negative rating indicated that their concerns had been addressed, although the rating remained negative in the system. The other reviewer with a negative rating did not engage in the discussion. However, the authors provided a rebuttal to the concerns. Overall, the paper received positive feedback from the reviewers, and the authors provided detailed rebuttals to each concern raised. Therefore, the paper is recommended for acceptance.

**Reviewer Concerns:**

The authors addressed most of the reviewers’ concerns, including the differences from existing works (GAN-based and diffusion-based approaches), the hyperparameters chosen during inference, clarification of the novelty, justification of the introduced dynamic modulation strategies, additional experimental results on different prompt styles, and the trade-off between enforced modulation and prompt fidelity.

**Reviewer Scores:**

Reviewers DbAo, Tihg, and rF5b participated in the discussion with the authors and indicated that their concerns had been addressed.

Reviewer Jrir did not participate in the discussion. The authors have provided a rebuttal to the two concerns raised by this reviewer, and the rebuttal contains adequate information addressing those concerns.

---

### Decision · Program_Chairs · 2026-01-26

Accept (Poster)